# The cell cycle regulator GpsB functions as cytosolic adaptor for multiple cell wall enzymes

Robert M. Cleverley[1], Zoe J. Rutter[1], Jeanine Rismondo[2,6], Federico Corona[3,7], Ho-Ching Tiffany Tsui [4], Fuad A. Alatawi[5], Richard A. Daniel [5], Sven Halbedel [2], Orietta Massidda [3,8], Malcolm E. Winkler [4] & Richard J. Lewis [1]

Bacterial growth and cell division requires precise spatiotemporal regulation of the synthesis and remodelling of the peptidoglycan layer that surrounds the cytoplasmic membrane. GpsB is a cytosolic protein that affects cell wall synthesis by binding cytoplasmic mini-domains of peptidoglycan synthases to ensure their correct subcellular localisation. Here, we describe critical structural features for the interaction of GpsB with peptidoglycan synthases from three bacterial species (*Bacillus subtilis*, *Listeria monocytogenes* and *Streptococcus pneumoniae*) and suggest their importance for cell wall growth and viability in *L. monocytogenes* and *S. pneumoniae*. We use these structural motifs to identify novel partners of GpsB in *B. subtilis* and extend the members of the GpsB interactome in all three bacterial species. Our results support that GpsB functions as an adaptor protein that mediates the interaction between membrane proteins, scaffolding proteins, signalling proteins and enzymes to generate larger protein complexes at specific sites in a bacterial cell cycle-dependent manner.

[1] Institute for Cell and Molecular Biosciences, University of Newcastle, Newcastle upon Tyne NE2 4HH, UK. [2] FG11 Division of Enteropathogenic Bacteria and Legionella, Robert Koch Institute, Burgstrasse 37, 38855 Wernigerode, Germany. [3] Dipartimento di Scienze Chirurgiche, Università di Cagliari, Cagliari 09100, Italy. [4] Department of Biology, Indiana University Bloomington, Bloomington, IN 47405, USA. [5] Centre for Bacterial Cell Biology, Institute for Cell and Molecular Biosciences, University of Newcastle, Newcastle upon Tyne NE2 4AX, UK. [6] Present address: Section of Microbiology and MRC Centre for Molecular Bacteriology and Infection, Imperial College London, London SW7 2DD, UK. [7] Present address: Centre for Bacterial Cell Biology, Institute for Cell and Molecular Biosciences, University of Newcastle, Newcastle upon Tyne NE2 4AX, UK. [8] Present address: Department CIBIO, University of Trento, via Sommarive 9, 38123 Povo, Italy. Correspondence and requests for materials should be addressed to R.J.L. (email: r.lewis@ncl.ac.uk)

Peptidoglycan (PG), a network of glycan strands connected by short peptides, forms the essential cell envelope that maintains cell shape and protects bacteria from osmotic stresses[1]. High molecular weight (HMW) bi-functional penicillin binding proteins (class A PBPs) are PG synthases that catalyse glycan strand polymerisation and peptide crosslinking, whereas HMW class B mono-functional PBPs only have transpeptidase functions[2]. The PG layer needs remodelling to enable normal cell growth and division and thus the bacterial cell cycle requires the extracellular activities of PBPs[3] and PG hydrolases[4] to be co-ordinated. The outer membrane-anchored LpoA/B lipoproteins activate their cognate PBP1A/1B PG synthases in the synthesis of the thin, periplasmic PG layer in the Gram-negative paradigm *Escherichia coli*[5,6]. By contrast, Gram-positive bacteria have a much thicker PG layer that is complemented with other anionic cell wall polymers. PG synthesis regulation in Gram-positive bacteria involves protein phosphorylation by orthologues of the serine/threonine kinase PknB[7]/StkP[8], and dedicated cell cycle scaffolding proteins including DivIVA[9], EzrA[10] and GpsB[11,12]. However, the molecular mechanisms that modulate PG synthesis in Gram-positive bacteria are virtually unknown.

GpsB has emerged as a major regulator of PG biosynthesis in low G+C Gram-positive bacteria, and its homologues (DivIVA/Wag31/antigen 84) in Actinobacteria play essential roles in hyphal growth and branching[13–15]. GpsB was initially characterised in *Bacillus subtilis* where severe cell division and growth defects were observed when both *gpsB* and *ezrA*[11] or *gpsB* and *ftsA*[12] were deleted. Both EzrA and FtsA play roles in the dynamics and membrane anchoring of the FtsZ Z-ring, the constriction of which is fundamental to cell division[16]. The Z-ring also recruits downstream proteins, including PBPs[17,18], to complete the process. Deletion of *gpsB* alone in *Listeria monocytogenes* caused marked growth and division defects at 37 °C and was lethal at 42 °C[19]. Moreover, *gpsB* deletion in *L. monocytogenes* also resulted in enhanced susceptibility to β-lactam[19] and fosfomycin[20] antibiotics, reduced virulence in an insect infection model[19], and caused alterations to PG structure[21]. Mutations in *gpsB* that affected binding to the PG synthase PBPA1 also showed a lethal phenotype in *L. monocytogenes* at 42 °C[19]. The *gpsB* gene is essential in the *Streptococcus pneumoniae* D39 progenitor strain as well as in some of its laboratory derivatives and its inactivation resulted in elongated cells unable to divide[22–24]. In addition, a genome-wide association study of *S. pneumoniae* clinical isolates revealed that the presence of *gpsB* variants was correlated significantly to β-lactam resistance[25], suggesting that GpsB may have fitness and pleiotropic roles in maintaining cell wall integrity during antibiotic stress. The *gpsB* gene has also been described as essential in the spherical bacterium *Staphylococcus aureus*, but the biochemical properties and physiological functions of *S. aureus* GpsB in this recent report[26] differ to what has been described in *B. subtilis*[11,12,18,19,27] *L. monocytogenes*[19,27] and *S. pneumoniae*[22–24].

In both *B. subtilis*[11] and *L. monocytogenes*[19], the cytosolic GpsB localises to the lateral side walls of newborn, growing cells and to the septum of dividing cells, the same localisation pattern as that of *B. subtilis* PBP1[11]. In *S. pneumoniae*, GpsB localises to mid-cell[22], the only region of active PG synthesis for both peripheral (side-wall) elongation and cell division in this bacterium. The localisation of GpsB at regions of active PG synthesis allows for the interaction of GpsB with the poorly characterised cytoplasmic mini-domains of PG synthases[11,19,27,28]. *S. pneumoniae* GpsB (*Sp*GpsB) has been found to co-immunoprecipitate with *Sp*PBP2a, *Sp*PBP2b and *Sp*MreC[24], suggesting these proteins form a complex that is regulated by *Sp*GpsB[24].

To gain molecular understanding of GpsB function, we report three crystal structures of PBP cytoplasmic mini-domains in complex with GpsB, the first structures of a PG synthase in complex with a cytoplasmic cell cycle regulator. Despite a marked absence of sequence and structural homology, we find that the PBP domains interact with equivalent surfaces in GpsB using an arginine that is conserved in the respective orthologues of the PBPs. The visualisation of each complex permits a comprehensive mutagenesis strategy and functional study to rationalise the role of each interfacial amino acid in the PBP:GpsB pairs. We uncover a sequence motif used by the *B. subtilis* PG synthase to interact with GpsB. We identify two new members of the GpsB interactome in this organism by querying the *B. subtilis* proteome with this motif for potential new partners of GpsB, and provide evidence for their connection to other, established proteins in growth and division. We also identify extensive GpsB interactomes in *B. subtilis*, *L. monocytogenes* and *S. pneumoniae* by bacterial two-hybrid assays (BACTH). Therefore, the role of GpsB in the bacterial cell cycle is that of an adaptor[29–31], docking PG synthases to other cell wall enzymes, scaffolds and shape determinants into protein complexes for division (the divisome) and peripheral growth (the elongasome).

## Results

**The first 16 residues of *Bs*PBP1 dictate the *Bs*GpsB interaction.** GpsB is an influential cell cycle regulator in low G+C Gram-positive bacteria and we set out to establish the common rules by which GpsB interacts with major PG synthases in three important bacteria—one model species (*B. subtilis*) and two pathogens (*L. monocytogenes* and *S. pneumoniae*). It has been determined previously by us by SPR that the first 31 amino acids of the cytoplasmic mini-domain of *B. subtilis* PBP1 (*Bs*PBP1) were critical for binding of *Bs*GpsB[19]. Using the same technique the binding site was further mapped to the first 16 amino acids of *Bs*PBP1 by comparing the interaction of *Bs*GpsB with SPR chips coated with full-length *Bs*PBP1 and a PBP1 deletion mutant, *Bs*PBP1$_{17-914}$, where the codons for the first 16 amino acids were removed genetically. While submicromolar concentrations of *Bs*GpsB bound to chips coated with the wild-type *Bs*PBP1, there was no interaction even when 25 μM *Bs*GpsB was injected over equivalent chips coated with *Bs*PBP1$_{17-914}$ (Fig. 1a).

We subsequently solved the crystal structure of the complex between the N-terminal domain of *Bs*GpsB whose termini were truncated slightly to expedite crystallisation, *Bs*GpsB$_{5-64}$, with the first 17 amino acids of *Bs*PBP1, *Bs*PBP1$_{1-17}$ (Fig. 1b, c). The *Bs*PBP1$_{1-17}$ α-helix is stabilised by an intramolecular salt bridge between *Bs*PBP1$_{1-17}$$^{Glu9}$ and *Bs*PBP1$_{1-17}$$^{Arg12}$ and by a hydrogen bond between the sidechain of *Bs*PBP1$_{1-17}$$^{Ser7}$ and the backbone amide of *Bs*PBP1$_{1-17}$$^{Ala10}$. A prominent feature of the complex is the deep penetration of the sidechain of *Bs*PBP1$^{Arg8}$ into the groove between *Bs*GpsB$_{5-64}$ α-helices 1 and 2, contacting the mainchain carbonyl oxygens of *Bs*GpsB$^{Ile13}$, *Bs*GpsB$^{Leu14}$ and *Bs*GpsB$^{Lys16}$ and forming a salt bridge with *Bs*GpsB$^{Asp31}$ (Fig. 1c), which in turn is tethered in place by hydrogen bonds to the hydroxyl of *Bs*GpsB$^{Tyr25}$. The backbone amides of *Bs*PBP1$^{Arg8}$ and *Bs*PBP1$^{Glu9}$ interact with *Bs*GpsB$^{Asp35}$, mimicking the mainchain interactions in successive turns in an α-helix. In a longer α-helix, the backbone amides of *Bs*PBP1$^{Arg8}$ and *Bs*PBP1$^{Glu9}$ would not be available to interact with *Bs*GpsB$^{Asp35}$ because of intra-helical hydrogen bonds with the mainchain carbonyls of *Bs*PBP1$^{Phe5}$ and *Bs*PBP1$^{Asn6}$. The sidechain of *Bs*PBP1$^{Arg11}$ forms hydrogen bonds with the carbonyl oxygen of *Bs*GpsB$^{Leu14}$ and a salt bridge with *Bs*GpsB$^{Glu17}$. Van der Waals' interactions connect *Bs*PBP1$^{Arg8}$ to *Bs*GpsB$^{Leu34}$ (Fig. 1c) and *Bs*PBP1$^{Glu9}$ with *Bs*GpsB$^{Lys32}$. The *Bs*PBP1$_{1-17}$ peptide binds to a groove between α-helices 1 and 2 in only one molecule of *Bs*GpsB$_{5-64}$ in the crystallographic asymmetric unit as the second

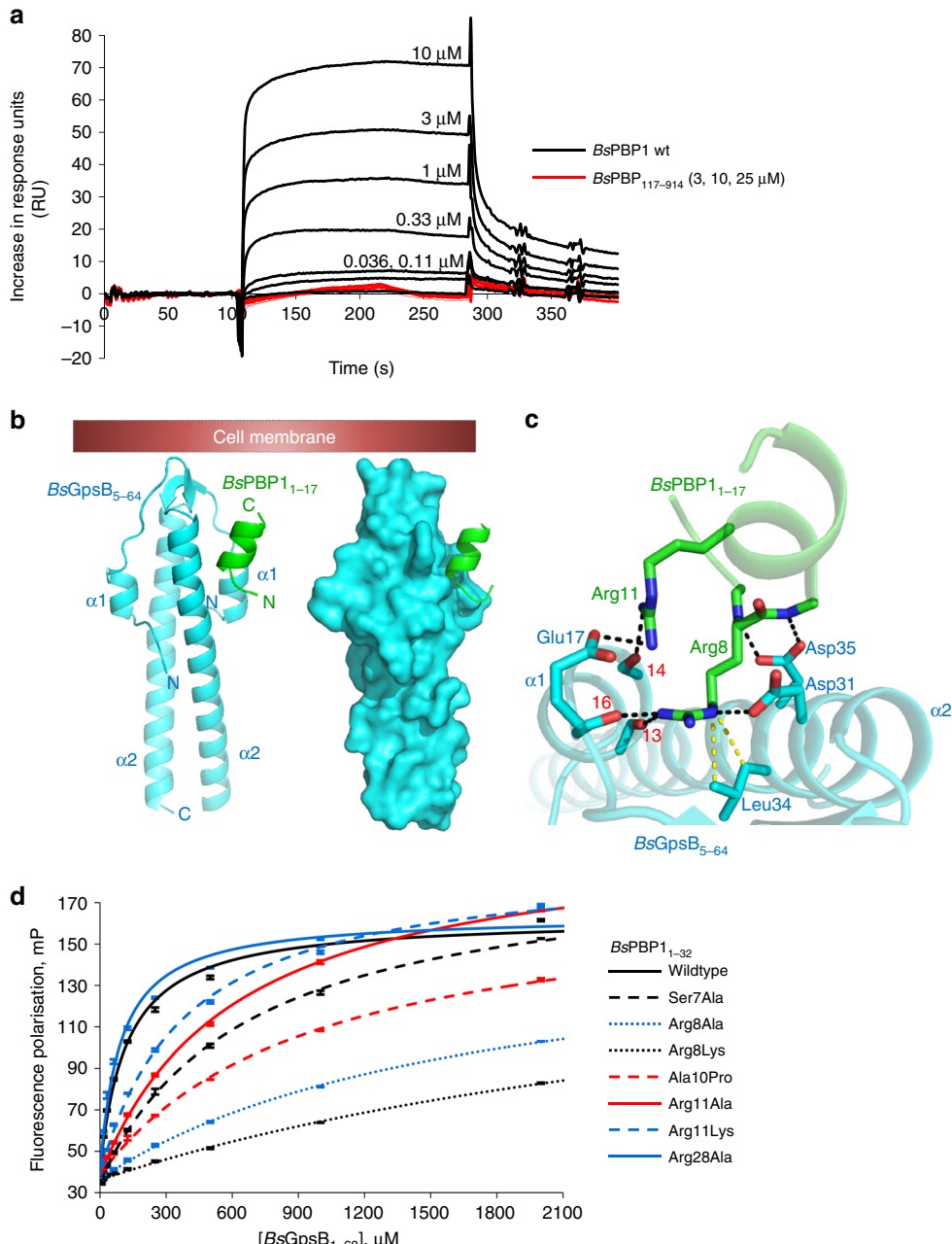

**Fig. 1** *Bs*GpsB:*Bs*PBP1 interactions require conserved arginines in the α-helical cytoplasmic minidomain of *Bs*PBP1. **a** *Bs*GpsB interacts with the first 16 amino acids of *Bs*PBP1. SPR sensorgrams of full-length *Bs*GpsB against immobilised full-length *Bs*PBP1 (black) and *Bs*PBP1$_{17-914}$ (red). *Bs*GpsB does not interact with the *Bs*PBP1$_{17-914}$ coated chip, even when 25 μM GpsB is injected. **b** Cartoon (left) and surface representation (right) of the crystal structure of the *Bs*GpsB$_{5-64}$:*Bs*PBP1$_{1-17}$ complex. *Bs*GpsB$_{5-64}$ is coloured cyan and *Bs*PBP1$_{1-17}$ is coloured green and the likely plane of the bacterial membrane is shown as a red box. **c** The *Bs*GpsB$_{5-64}$:*Bs*PBP1$_{1-17}$ complex is dependent upon a conserved SRxxR(R/K) motif in *Bs*PBP1. Key interfacial residues in the *Bs*GpsB$_{5-64}$:*Bs*PBP1$_{1-17}$ complex are shown as sticks and coloured (and labelled) blue and green, respectively. The carbonyl oxygens of *Bs*GpsB$^{Ile13}$, *Bs*GpsB$^{Leu14}$ and *Bs*GpsB$^{Lys16}$ are labelled with their respective red numerals. Hydrogen bonds are shown as black dashed lines and the van der Waals' interactions between *Bs*GpsB$^{Leu34}$ and *Bs*PBP1$^{Arg8}$ are in yellow. **d** Mutation of key *Bs*PBP1 interfacial residues in the structure of *Bs*GpsB$_{5-64}$:*Bs*PBP1$_{1-17}$ leads to a loss of binding of position 16 TAMRA-labelled *Bs*PBP1$_{1-32}$ variants to *Bs*GpsB$_{1-68}$ as measured by fluorescence polarisation. When the *Bs*PBP1$_{1-32}$ peptide was labelled with fluorescein at position 31 the measured affinity was the same as if the fluorophore was at position 16, indicating that the fluorophore itself or its position in the peptide has no impact on the binding interaction. The calculated dissociation constants can be found in Supplementary Table 1

*Bs*GpsB-binding site is blocked by crystal contacts. The key interactions are mapped onto GpsB and PBP1 sequences in Supplementary Figure 1A.

To rationalise the features of PBP cytoplasmic domains important for determining recognition by GpsB, the interactions described above were analysed further by fluorescence polarisation

(FP) and circular dichroism (CD). The *Bs*GpsB$_{1-68}$$^{Glu17Ala}$, *Bs*GpsB$^{Tyr25Phe}$, *Bs*GpsB$_{1-68}$$^{Asp31Ala}$ and *Bs*GpsB$_{1-68}$$^{Asp35Ala}$ mutations had little impact on protein stability (Supplementary Figure 1B and Supplementary Note 1) and each reduced the affinity for *Bs*PBP1$_{1-32}$ by more than 8-fold (Supplementary Figure 1C, Supplementary Table 1). *Bs*PBP1$^{Arg8Lys}$,

$Bs$PBP1$^{Arg8Ala}$ and $Bs$PBP1$^{Arg11Ala}$ mutations each resulted in reduced affinities for $Bs$GpsB$_{1-68}$ by at least 5-fold (Fig. 1d, Supplementary Table 1). $Bs$PBP1$_{1-32}$$^{Arg28Ala}$ had no effect on binding (Fig. 1d, Supplementary Table 1), suggesting that the overall positive charge of the peptide is not the primary determinant of $Bs$GpsB:$Bs$PBP1 interactions, rather the unique physicochemical characteristics of an arginine at position 8 in $Bs$PBP1 is essential. The $Bs$PBP1$_{1-32}$$^{Ser7Ala}$ and $Bs$PBP1$_{1-32}$$^{Ala10Pro}$ mutations each reduced the affinity for $Bs$GpsB$_{1-68}$ by at least 6-fold by affecting the α-helix of BsPBP11-32 (Fig. 1d, Supplementary Figure 1D, Supplementary Table 1 and Supplementary Note 1), and the reduction in affinity was the same irrespective of the position of the fluorophore in the peptide (Supplementary Figure 1E, Supplementary Table 1). $Bs$PBP1$_{1-32}$$^{Ser7}$ acts as the helix N-cap, a role that is performed preferentially by Ser, Asn and Thr[32], and alanine and proline substitutions in helical positions equivalent to $Bs$PBP1$_{1-32}$$^{Ser7}$ and $Bs$PBP1$_{1-32}$$^{Ala10}$, respectively, destabilise model peptides[32,33]. Finally, the importance of $Bs$PBP1$_{1-32}$$^{Ser7}$, $Bs$PBP1$_{1-32}$$^{Arg8}$ and $Bs$PBP1$_{1-32}$$^{Arg11}$ to GpsB binding is highlighted because these are the most well conserved residues in an alignment of the cytoplasmic mini-domains of $Bacillaceae$ PBP1 PG synthases (Supplementary Figure 1F).

**$L.$ monocytogenes GpsB interacts with PBPA1 by a TRxxYR motif.** The deletion of $gpsB$ alone in $B.$ subtilis has no readily apparent phenotype until combined with deletions in $ezrA$[11] or $ftsA$[12]; by contrast, the deletion of $gpsB$ in $L.$ monocytogenes is lethal when grown at 42 °C[19]. Since GpsB in both species interact with class A PG synthases, we next determined whether the rules established above for the $Bs$GpsB:$Bs$PBP1 interaction could be applied directly to $Lm$GpsB:$Lm$PBPA1. The cytoplasmic mini-domain of $Lm$PBPA1, $Lm$PBPA1$_{1-29}$, has an abundance of positively charged residues, but lacks an exact copy of the SRxxR(R/K) motif of $Bacillaceae$ PBP1 (Supplementary Figure 1F, 2A); the closest equivalent is TRxxYR. In FP, the first 20 amino acids of $Lm$PBPA1, $Lm$PBPA1$_{1-20}$, bound to the N-terminal domain of $Lm$GpsB, $Lm$GpsB$_{1-73}$, with an affinity similar to that of $Bs$PBPA1$_{1-32}$ for $Bs$GpsB$_{1-68}$ (Supplementary Figure 2B, Supplementary Table 1), but we were unable to co-crystallise $Lm$GpsB constructs with $Lm$PBPA1 peptides to visualise these interactions and to compare them to $Bs$GpsB$_{5-64}$:$Bs$PBP1$_{1-17}$. Consequently, and to expedite crystallisation, we solved the structure of the first 15 amino acids of $Lm$PBPA1, $Lm$PBPA1$_{1-15}$, bound to $Bs$GpsB$_{5-64}$$^{Lys32Glu}$, which is a surrogate for $Lm$GpsB$_{1-73}$ because (i) all the residues within 8 Å of the $Bs$GpsB$_{5-64}$:$Bs$PBP1$_{1-17}$ interface are conserved in $Lm$GpsB except for Lys32, which is glutamate in $Lm$GpsB, and thus all peptide-contacting residues are maintained—as well as bystander residues that help indirectly to shape the PBP binding site; (ii) $Lm$GpsB and $Bs$GpsB use over-lapping PBP binding sites[19]; (iii) the $K_d$s of $Bs$GpsB$_{1-68}$$^{Lys32Glu}$ and $Lm$GpsB$_{1-73}$ for the first 20 amino acids of $Lm$PBPA1, $Lm$PBPA1$_{1-20}$, are almost identical (Supplementary Figure 2B, Supplementary Table 1). Therefore, the K32E variant of the N-terminal domain of $Bs$GpsB is as close a surrogate for the equivalent $Lm$GpsB domain that could be obtained.

The subsequent structure of the $Bs$GpsB$_{5-64}$$^{Lys32Glu}$:$Lm$PBPA1$_{1-15}$ complex revealed that, in contrast to the $Bs$GpsB$_{5-64}$:$Bs$PBP1$_{1-17}$ complex, the majority of the $Lm$PBPA1 peptide was disordered except for $Lm$PBPA1$_{1-15}$$^{Arg8}$ and the immediately-adjacent main-chain atoms. $Lm$PBPA1$_{1-15}$$^{Arg8}$ adopts the same orientation and makes the same interactions as described above for $Bs$PBP1$_{1-17}$$^{Arg8}$ in the $Bs$GpsB$_{5-64}$:$Bs$PBP1$_{1-17}$ complex (Figs. 1c, 2a). The bidentate

interaction of Glu35 of $Bs$GpsB$_{5-64}$$^{Lys32Glu}$ with the backbone amides of $Lm$PBPA1$^{Arg8}$ and $Lm$PBPA1$^{Ser9}$ are maintained just as in the $Bs$GpsB$_{5-64}$:$Bs$PBP1$_{1-17}$ complex and, as the backbone torsion angles of $Lm$PBPA1$_{1-15}$$^{Arg8}$ are also α-helical, it suggests that the role of the conserved $Lm$PBPA1$^{Thr7}$ is to N-cap this helix[32]. The interaction of $Bs$GpsB$_{5-64}$$^{Lys32Glu}$ with $Lm$PBPA1$_{1-15}$ seemingly centres almost entirely on a single arginine. How $Lm$GpsB discerns $Lm$PBPA1$^{Arg8}$ over other positively charged residues in the arginine- and lysine-rich cytoplasmic domain of $Lm$PBPA1 (Supplementary Figure 2A) was determined by FP and BACTH. $Lm$PBPA1$_{1-20}$$^{Arg8Ala}$ and $Lm$PBPA1$_{1-20}$$^{Arg12Ala}$ mutations reduced the affinity for $Bs$GpsB$_{1-68}$$^{Lys32Glu}$ by >15- and ~4-fold, respectively (Supplementary Figure 2B, Supplementary Table 1), and had significant negative impact on the $Lm$GpsB:$Lm$PBPA1 interaction by BACTH (Fig. 2b). Alanine substitution of $Lm$PBPA1$^{Tyr11}$ had a comparatively milder impact in BACTH (Fig. 2b) and in FP (Supplementary Figure 2B, Supplementary Table 1). Reintroducing positive charge into $Lm$PBPA1$_{1-20}$$^{Arg8Ala}$ did not restore wild-type binding affinity in FP as $Lm$PBPA1$_{1-20}$$^{Arg8Ala,Ser16Arg}$ still bound to $Bs$GpsB$_{1-68}$$^{Lys32Glu}$ with an affinity at least 10-fold weaker than wild-type (Supplementary Figure 2B, Supplementary Table 1). BACTH also supports the central importance of $Lm$PBPA1$^{Thr7}$, which presumably plays a structural role in the $Lm$PBPA1$_{1-15}$ peptide possibly by acting as an N-cap and positioning $Lm$PBPA1$^{Arg8}$ at the start of an α-helix at the. Alanine substitution of the other positively charged residues downstream of residue 14 had no impact in BACTH (Fig. 2b), consistent with the particularly stringent specificity of $Lm$GpsB for $Lm$PBPA1$^{Arg8}$ in comparison to other positively charged residues.

$Lm$PBPA1$^{Arg8}$ and $Bs$PBP1$^{Arg8}$ are equivalent in their interactions with GpsB. Of the other GpsB-binding determinants of $Bs$PBP1, $Lm$PBPA1 lacks an analogous $Bs$PBP1$^{Arg11}$. The sequential equivalent is $Lm$PBPA1$^{Tyr11}$, but this residue is completely disordered and its mutation to alanine reduced the affinity for $Bs$GpsB$_{1-68}$$^{Lys32Glu}$ only by 2-fold (Supplementary Figure 2B). To further decipher the reason for the specificity for $Lm$PBPA1$^{Arg8}$, the importance of α-helix formation in $Lm$PBPA1$_{1-15}$ for GpsB binding was analysed with Q10P-mutated $Lm$PBPA1 peptides. A Q10P mutation caused a >7-fold reduction in binding affinity (Supplementary Figure 2B, Supplementary Table 1) and CD confirmed a significant impact of the Q10P mutation on the α-helicity of $Lm$PBPA1$_{1-15}$ peptides (Supplementary Figure 2C). Finally, the effects of mutations to the crucial $Lm$GpsB-interacting residues in $Lm$PBPA1 were also probed in $Listeria$ using fosfomycin sensitivity as a reporter since $\Delta gpsB$ mutants are more susceptible to this antibiotic at 37 °C than wild-type $L.$ monocytogenes[20]. Fosfomycin inhibits the first enzyme in the biosynthetic pathway of PG, MurA, and the $\Delta gpsB$ mutant is hypersensitive to it probably because of unproductive consumption of PG precursors due to mis-regulated $Lm$PBPA1[20]. By contrast, removal of $pbpA1$ reduces substrate turnover in PG biosynthesis and thus the cells become more resistant to fosfomycin, and a $pbpA1$ $pbpA2$ double mutant is not viable[34]. Effects on fosfomycin sensitivity were apparent in mutants carrying the $pbpA1$$^{Arg8Ala,Arg12Ala}$ and $pbpA1$$^{Gln10Pro}$ alleles but only when $Lm$PBPA2, the $Lm$PBPA1 paralogue, was also absent (Fig. 2c). Synthetic lethality with $pbpA2$ and a growth defect at 42 °C is characteristic of the $L.$ monocytogenes null $gpsB$ mutant[19], suggesting that the observed effects partially phenocopy $\Delta gpsB$. However, no $pbpA1$ mutation phenocopied the $\Delta gpsB$ mutant completely (Supplementary Figure 2D).

Taken together, our data highlight the importance of a conserved arginine and α-helicity in class A PG synthases for interacting with GpsB in two species. In both cases the arginine is adjacent to a conserved residue with high propensity to act as a

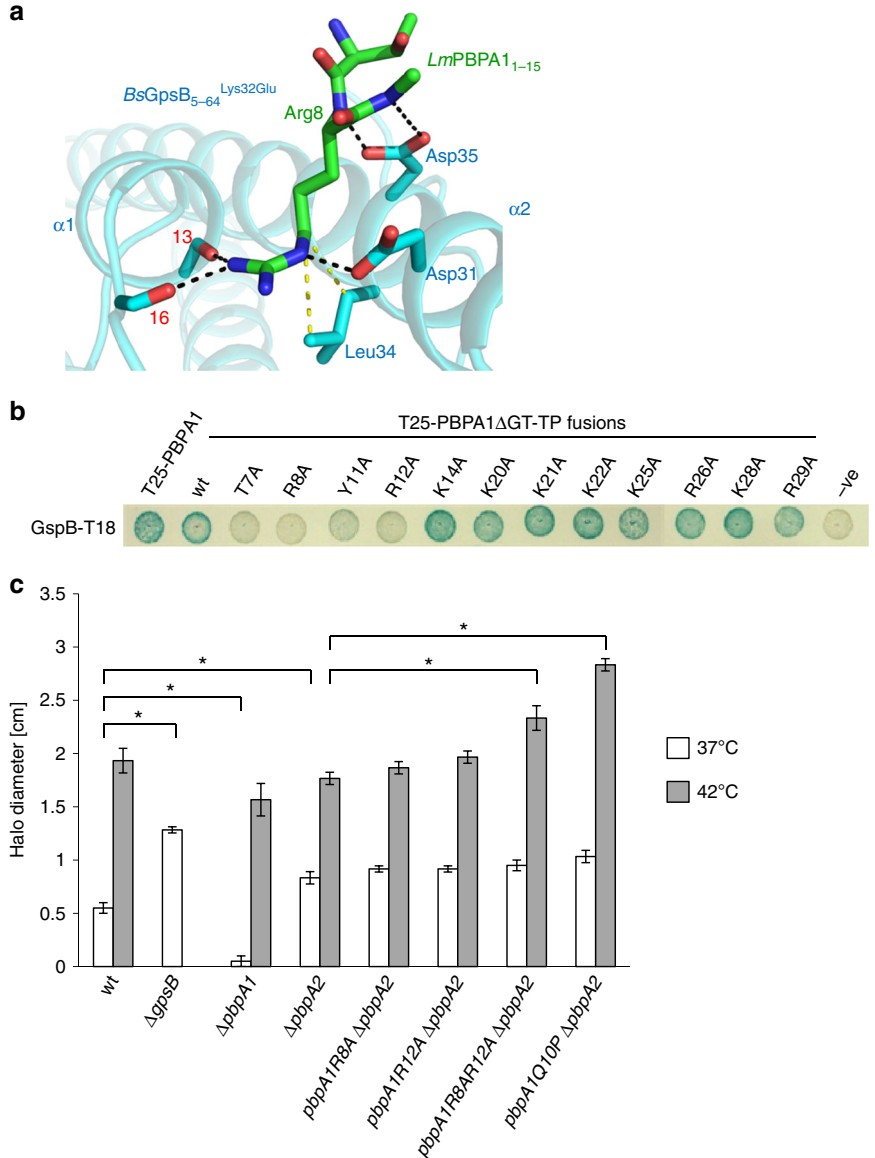

**Fig. 2** The *Lm*GpsB:*Lm*PBPA1 interactions are also governed by a conserved arginine. **a** The structure of the *Bs*GpsB$_{5-64}^{Lys32Glu}$:*Lm*PBPA1$_{1-15}$ complex reveals that only Arg8 and adjacent backbone atoms of the *Lm*PBPA1$_{1-15}$ peptide are ordered. In this cartoon, *Bs*GpsB$_{5-64}^{Lys32Glu}$ is coloured cyan and selected sidechains are drawn as stick with cyan carbons, whereas the *Lm*PBPA1$_{1-15}$ peptide is represented in stick form, with green carbons. The carbonyl oxygens of *Bs*GpsB$^{Ile13}$ and *Bs*GpsB$^{Lys16}$ are denoted by respective red numerals. Hydrogen bonds are shown as black dashed lines and the van der Waals' interactions between *Bs*GpsB$^{Leu34}$ and *Lm*PBPA1$^{Arg8}$ are in yellow. Only one PBP-binding site is occupied by peptide in these crystals because the second site is blocked by crystal contacts. **b** Mutation of conserved *Lm*PBPA1 residues results in a loss of interaction by BACTH. The removal of residues 92–827, correlating to the glycosyltransferase and transpeptidase domains of *Lm*PBPA1, results in the PBPA1ΔGT-TP peptide. Empty pKT25 (−) was used as a negative control. Agar plates were photographed after 24 h at 30 °C. **c** Effect of N-terminal *pbpA1* mutations on fosfomycin sensitivity of a Δ*pbpA2* mutant. Wild-type and mutant *L. monocytogenes* EGD-e strains were grown as confluent layers on BHI agar plates at 37 °C and 42 °C and halo diameters around fosfomycin-containing filter discs were measured and corrected for the disc diameter. The experiment was performed in triplicate, and average values and standard deviations are shown. Asterisks indicate statistically significant differences (*P* < 0.01, *t*-test)

helix N-cap, implying that positioning of the arginine at the start of the helix is critical. Furthermore, since *pbpA1* does not phenocopy *gpsB* in *L. monocytogenes*, and *gpsB* deletion on its own in *B. subtilis* has no clear phenotype, GpsB must have additional functions in both bacteria.

**Extending the *B. subtilis* and *L. monocytogenes* GpsB interactomes**. The data presented above describe features critical for interactions involving *Bs*GpsB, which include a helical SRxxR(R/K) motif in close proximity to the membrane. To identify hitherto unidentified *Bs*GpsB-interacting proteins, the *B. subtilis* proteome was queried with the SRxxR(R/K) motif. Two previously uncharacterised ORFs, *Bs*YpbE and *Bs*YrrS, conform to all the features described above. *Bs*YpbE is a membrane protein with a 56-residue cytoplasmic domain that encodes a SRVERR motif. The extracellular region, residues 80–240, contains a LysM (lysin motif) domain between residues 189 and 235; LysM domains are ~40-residue, degenerate PG- and chitin-binding modules widespread in bacteria and eukaryotes. *yrrS* is found in a bicistronic operon widely conserved in the *Bacillaceae* with the gene (*yrrR*)

encoding a class B PBP, PBP4b[35], suggesting these genes have a linked function in cell wall homoeostasis[36]. $Bs$YrrS comprises an 18-residue cytoplasmic domain with two potential, overlapping $Bs$GpsB-binding motifs SRYENR and NRDKRR and an extracellular domain that belongs to the widespread and currently uncharacterised DUF1510 family.

LysM domains are frequently found as tandem repeats within bacterial proteins[37] and the individual domains can act co-operatively to bind PG[38,39]. $Bs$YpbE contains one LysM domain hence oligomerization of $Bs$YpbE may enhance PG binding, with the oligomerisation of the extracellular LysM domain of $Bs$YpbE presumably controlled by cytoplasmic, hexameric $Bs$GpsB[27], the essential form of the protein in bacteria[19]. In the absence of purified, full-length $Bs$YpbE to test this hypothesis directly, monomeric and dimeric forms of $Bs$YpbE$_{130-240}$, which encompasses the sole extracellular LysM domain, were generated instead. Dimeric $Bs$YpbE$_{130-240}$ was prepared by mutation of Ser132 to cysteine to enable disulphide-linked $Bs$YpbE$_{130-240}$$^{Ser132Cys}$ dimers to be purified. In pulldown assays, the binding of $Bs$YpbE$_{130-240}$$^{Ser132Cys}$ dimers to PG was enhanced considerably relative to the monomeric, cysteine-free version of $Bs$YpbE$_{130-240}$ (Supplementary Figure 3A). Therefore, the binding of YpbE to PG in $B.$ $subtilis$ will be stimulated by YpbE multimerisation, induced by hexameric GpsB.

The interaction of $Bs$GpsB$_{1-68}$ with $Bs$YrrS and $Bs$YpbE was assessed by FP and BACTH. $Bs$GpsB$_{1-68}$ bound to a 21-residue fragment of the cytoplasmic domain of $Bs$YpbE, $Bs$YpbE$_{1-21}$ that encompasses the SRVERR motif, and the entire cytoplasmic mini-domain of $Bs$YrrS, $Bs$YrrS$_{1-18}$, with $K_d$ values of 13 μM (Fig. 3a) and 430 μM (Fig. 3b), respectively. The modest affinity of the $Bs$YrrS$_{1-18}$ peptide for $Bs$GpsB$_{1-68}$ (Supplementary Table 1) probably translates to a substantially tighter affinity in bacterial cells, because of an avidity effect resulting from $Bs$GpsB and $Bs$YrrS associating with membranes, effectively increasing the local concentration of each significantly. The specificity of these interactions measured by FP was consistent with the impact of $Bs$GpsB$^{Asp31Ala}$ and $Bs$GpsB$^{Tyr25Phe}$ mutations, each of which reduced the affinities for $Bs$YrrS$_{1-18}$ and $Bs$YpbE$_{1-21}$ by 7- and ~40-fold, respectively (Fig. 3a, b), and in-line with the roles of $Bs$GpsB$^{Asp31}$ and $Bs$GpsB$^{Tyr25}$ in defining the $Bs$PBP1 binding site. Interactions of $Bs$GpsB with $Bs$YrrS and $Bs$YpbE were also detected by BACTH, with the interactions mapping to the N-terminal domain of $Bs$GpsB in both cases (Fig. 3c). GpsB is only conditionally essential in $B.$ $subtilis$[11,12], and perhaps it is not surprising that no obvious cell growth or division phenotypes were identified in our hands or previously[40] on deleting the genes encoding $Bs$YrrS, $Bs$PBP4b or $Bs$YpbE. BACTH was used to confirm that $Bs$YrrS interacted with $Bs$PBP4b, $Bs$PBP1 and $Bs$RodZ; the latter two proteins have established roles in cell division, growth and morphogenesis[41,42]. The interaction between $Bs$PBP1 and a fragment of $Bs$YrrS lacking residues 13–16, $Bs$YrrS$_{\Delta13-16}$ (which reduced non-specific binding to the $Bs$PBP1-immobilised SPR chip and did not affect the SRYENR motif) was quantified by SPR, and $Bs$YrrS$_{\Delta13-16}$ bound to $Bs$PBP1 with a $K_d$ of 20 nM (Supplementary Figure 3B). Therefore, these gene products are capable of forming a network of interactions (Fig. 3d) that may be nucleated by the formation of a $Bs$PBP1: $Bs$YrrS complex.

Homologues of YpbE do not exist in $L.$ $monocytogenes$ and the YrrS homologue (Lmo1495) does not contain a signature $Bs$GpsB-binding motif and neither protein is found in $S.$ $pneumoniae$. No strong potential GpsB-interacting candidates were identified when the $L.$ $monocytogenes$ proteome was searched with either TRxxYR or SRxxR(R/K) as the query. BACTH was thus used to uncover additional potential $Lm$GpsB functions in $L.$ $monocytogenes$ using a bank of 27 listerial

components from the known elongation and division machineries in bacteria (Supplementary Note 2). Twelve proteins were shown to interact with $Lm$GpsB and only these are shown in Supplementary Figure 3C. There is no consensus motif shared by these proteins, though all have at least one arginine present in their cytoplasmic regions that is conserved in their respective orthologues. Two classes of hits were identified in the BACTH screen; class I hits ($Lm$PBPA1, $Lm$MreC and $Lm$SepF, and $Lm$GpsB self-interactions) turned blue after one day of incubation (Supplementary Figure 3C). Class II hits turned blue after 2 days incubation at 30 ℃, including $Lm$ZapA, $Lm$EzrA, $Lm$DivIB, $Lm$DivIC, $Lm$MreC, $Lm$MreBH and the other HMW $Lm$PBPs (Supplementary Figure 3C). All of these interactions, except for the GpsB self-interactions, required the $Lm$GpsB N-terminal domain (Supplementary Figure 3C). In good agreement with the absence of a TRxxYR motif in $Lm$MreC, $Lm$SepF and $Lm$ZapA, interactions with these proteins did not require key residues in the known PBP-binding groove in $Lm$GpsB (Supplementary Figure 3D) and reciprocal tests validated the $Lm$GpsB:$Lm$PBPA1 interactions[21]. It would thus seem that $Lm$PBPA1 represents the only GpsB binding partner that employs the TRxxYR motif in $L.$ $monocytogenes$.

**Two arginines dictate S. pneumoniae GpsB:PBP2a molecular recognition.** $S.$ $pneumoniae$, more distantly related to either $Bacillus$ or $Listeria$, is an ovoid-shaped Gram-positive coccus in which GpsB is essential[22–24]. $Sp$GpsB co-immunoprecipitated with $Sp$PBP2a (one of three pneumococcal class A PBPs), $Sp$MreC and other proteins, suggesting they interact at some point in the pneumococcal cell cycle[24]. Synthetic lethality studies in pneumococcal Δ$gpsB$ suppressor mutants revealed that $pbp1a$, and not $pbp2a$, became essential in the absence of $gpsB$ indicating that $Sp$PBP2a is the class A PBP regulated by $Sp$GpsB in $S.$ $pneumoniae$[24]. We found that the cytoplasmic mini-domain of $Sp$PBP2a and many of its orthologues contain the consensus sequence (S/R)RS(R/G)(K/S)xR (Supplementary Figure 4A) that resembles the $Bacillaceae$ PBP1 SRxxR(R/K) motif (Supplementary Figure 1F). A 22-residue peptide of $Sp$PBP2a that encompasses this region, $Sp$PBP2a$_{23-45}$, was found by FP to bind to the N-terminal domain of $Sp$GpsB, $Sp$GpsB$_{1-63}$, with a $K_d$ of 80 μM whereas $Sp$GpsB$_{1-63}$$^{Asp33Ala}$ (equivalent to $Bs$GpsB$^{Asp35Ala}$) had a ~40-fold reduced affinity for $Sp$PBP2a$_{23-45}$ (Fig. 4a, Supplementary Table 1). The crystal structure of a slightly truncated form of the N-terminal domain of $Sp$GpsB, $Sp$GpsB$_{4-63}$ (to expedite crystallisation), was solved in the presence of a 14-residue peptide of $Sp$PBP2a, $Sp$PBP2a$_{27-40}$, which includes the (S/R)RS(R/G)(K/S)xR motif. In this instance, each subunit of the $Sp$GpsB dimer is peptide-bound (Fig. 4b). Peptide binding principally involves two arginines but each $Sp$GpsB subunit recognises the peptide differently. In $Sp$GpsB$_{4-63}$ molecule 1, $Sp$PBP2a$_{27-40}$ recognition centres on $Sp$PBP2a$^{Arg31}$ and $Sp$PBP2a$^{Arg36}$ (Fig. 4c), whereas molecule 2 involves $Sp$PBP2a$^{Arg33}$ and $Sp$PBP2a$^{Arg36}$ (Fig. 4d). The arginine pairs occupy the same positions as $Bs$PBP1$^{Arg11}$ and $Bs$PBP1$^{Arg8}$ in the $Bs$GpsB$_{5-64}$:$Bs$PBP1$_{1-17}$ complex (Fig. 1c); $Sp$PBP2a$^{Arg36}$ is equivalent to $Bs$PBP1$^{Arg11}$whereas $Sp$PBP2a$^{Arg31}$ and $Sp$PBP2a$^{Arg33}$ are equivalent to $Bs$PBP1$^{Arg8}$.

The $Sp$GpsB:$Sp$PBP2a interaction was confirmed by BACTH (Supplementary Figure 4B). The interaction was lost completely with $Sp$GpsB$^{Tyr23Ala}$, $Sp$GpsB$^{Val28Ala}$, $Sp$GpsB$^{Asp29Ala}$, $Sp$GpsB$^{Leu32Ala}$ and $Sp$GpsB$^{Asp33Ala}$ mutated proteins and reduced with $Sp$GpsB$^{Ile36Ala}$ (Fig. 5a). All the $Sp$GpsB variants retained the ability to self-interact and to interact with wild-type $Sp$GpsB (Fig. 5a, Supplementary Figure 4C); the impact of the mutations on interactions with $Sp$PBP2a thus does not reflect impaired expression of the relevant fusion proteins. Moreover, all

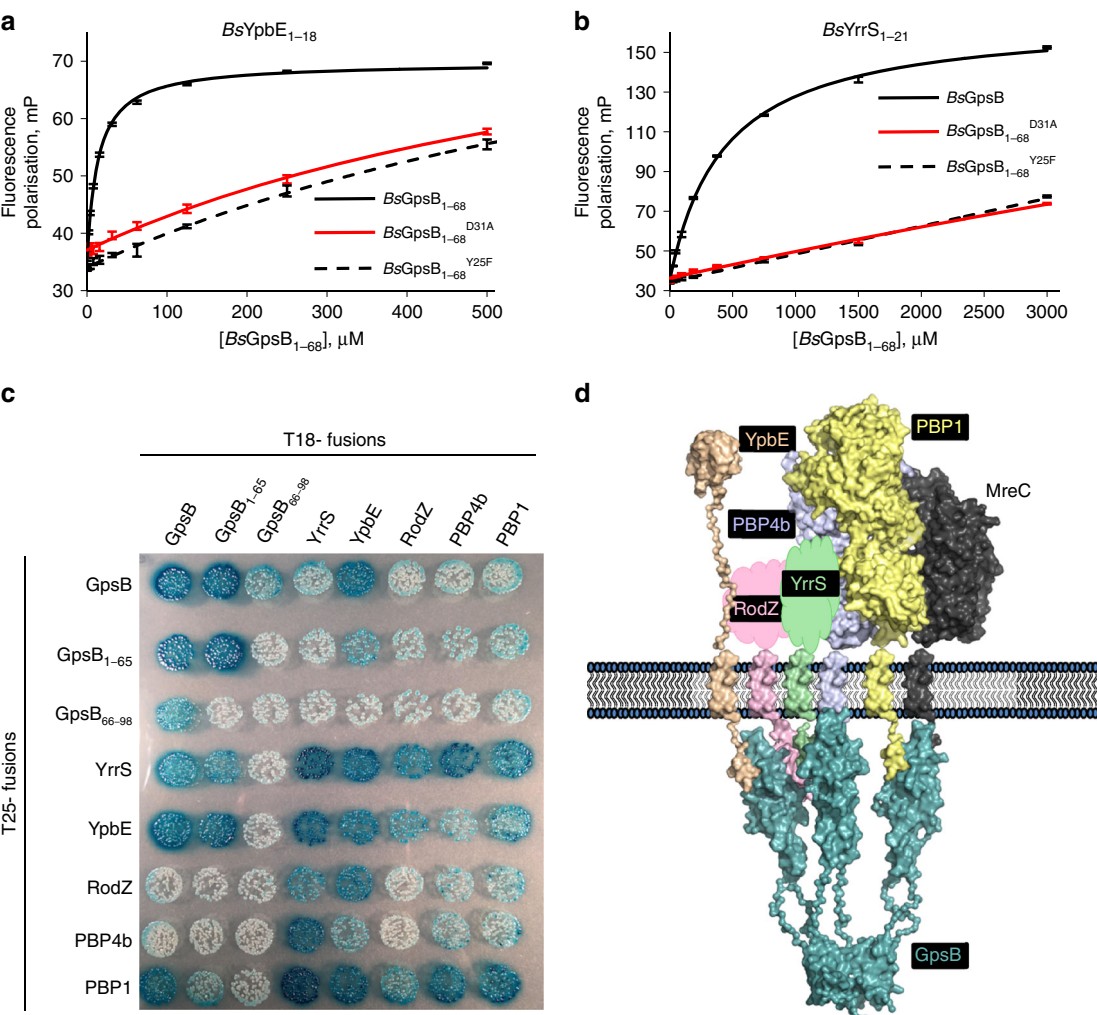

**Fig. 3** The SRxxR(R/K) motif identifies *Bs*YpbE and *Bs*YrrS as new *Bs*GpsB binding partners. **a**, **b** *Bs*YpbE$_{1-18}$ and *Bs*YrrS$_{1-21}$ bind to *Bs*GpsB$_{1-68}$ at the same site as *Bs*PBP1. Fluorescence polarisation of *Bs*GpsB$_{1-68}$ binding to fluorescein-labelled *Bs*YpbE$_{1-18}$ (**a**) and fluorescein-labelled *Bs*YrrS$_{1-21}$ (**b**). The interaction of wild-type proteins is depicted by black curves, whereas the red curves and dashed black lines correspond to *Bs*GpsB$_{1-68}$$^{Asp31Ala}$ and *Bs*GpsB$_{1-68}$$^{Tyr25Phe}$ mutants, respectively. **c** BACTH reveals a new *Bs*GpsB interaction network involving proteins encoding the SRxxR(R/K) motif. The panel shows pairwise combinations of proteins expressed as N-terminal fusions to both halves of adenylate cyclase in the BACTH host strain. Their presence in complexes containing *Bs*RodZ, *Bs*PBP4b and *Bs*PBP1 imply roles for *Bs*YrrS and *Bs*YpbE in sidewall synthesis during cell growth. The validity of the observed interactions is supported by the behaviour of the T18-GpsB$_{66-98}$ fusion, which does not interact with any other partner except T25-GpsB, and therefore acts as an internal control. The T18-*Bs*GpsB$_{66-98}$:T25-*Bs*GpsB interaction is consistent with the hexameric nature of *Bs*GpsB[19,26]. The absence of interactions between *Bs*GpsB$_{66-98}$ and *Bs*YrrS or *Bs*YpbE is consistent with data in **b** that shows that the N-terminal domain of GpsB interacts with the SRxxR(R/K) motif of YpbE and YrrS. **d** A model recapitulates interactions between *Bs*GpsB and partners. Surface representations of the SAXS structure of *Lm*GpsB[26] (coloured teal) and the closest homologues in the PDB by sequence (*Bs*PBP1, yellow, PDBid 2OLV; *Bs*PBP4b, pale blue, 4L0L; *Bs*RodZ, pink, 2WUS; *Bs*MreC, dark grey, 2J5U; *Bs*YpbE, salmon, 2MKX and the linker represents the disordered region, residues 131-188) are used as models for the *B. subtilis* proteins. Amorphous blobs, the surface area of which are scaled proportional to molecular weight, are used for *Bs*YrrS (green) and the extracellular domain of *Bs*RodZ (pink) where there is no structural information. The N-terminal domain of each membrane protein is cytoplasmic and 22-amino acid model helices represent each TM helix. The TMpred-predicted TM boundaries are: *Bs*PBP1 (38-60); *Bs*PBP4b (9-31); *Bs*YrrS (19-41); *Bs*YpbE (57-79); *Bs*RodZ (89-111) and *Bs*MreC (7-24). The GpsB-interacting domains of *Bs*YpbE, *Bs*YrrS and *Bs*PBP1 are based on the *Bs*PBP1$_{1-17}$ structure

the *Sp*GpsB variants, except *Sp*GpsB$^{Asp29Ala}$, retained some ability to interact with *Sp*MreC, which was also confirmed to interact with *Sp*GpsB by BACTH (Fig. 5a, Supplementary Figure 4B).

Despite differences in the secondary structures of the two independent *Sp*PBP2a peptides bound to the *Sp*GpsB$_{4-63}$ dimer (Fig. 4b), the two arginines form a similar network of interactions with *Sp*GpsB as described for *Bs*GpsB and *Lm*GpsB (Figs. 1c, 2a) with additional sidechain contacts in molecule 1 between *Sp*PBP2a$^{Ser32}$ and *Sp*GpsB$^{Asp33}$ (Fig. 4c), and *Sp*PBP2a$^{Arg31}$ and *Sp*GpsB$^{Tyr23}$. The importance of *Sp*PBP2a$^{Arg31}$ and *Sp*PBP2a$^{Arg33}$ is further supported by their sequence conservation

(Supplementary Figure 4A) and FP (Fig. 4a, Supplementary Table 1). Although *Sp*PBP2a$^{Arg31Lys}$ had only a 2-fold reduced affinity, which probably reflects the ability of *Sp*PBP2a$^{Arg33}$ to compensate for the loss of *Sp*PBP2a$^{Arg31}$, the binding affinity of *Sp*PBP2a$^{Arg31Lys,Arg33Lys}$ was reduced >25-fold relative to wild type. The importance of the *Sp*GpsB residues involved in the interactions with *Sp*PBP2a is also consistent with the phenotype because of the severe growth (Fig. 5b) and morphological defects (Fig. 5c) of *S. pneumoniae* strains harbouring the *Sp*GpsB$^{Tyr23Ala}$, *Sp*GpsB$^{Val28Ala}$, *Sp*GpsB$^{Asp29Ala}$, *Sp*GpsB$^{Leu32Ala}$ and *Sp*GpsB$^{As-p33Ala}$ alleles even though the mutated proteins were still capable

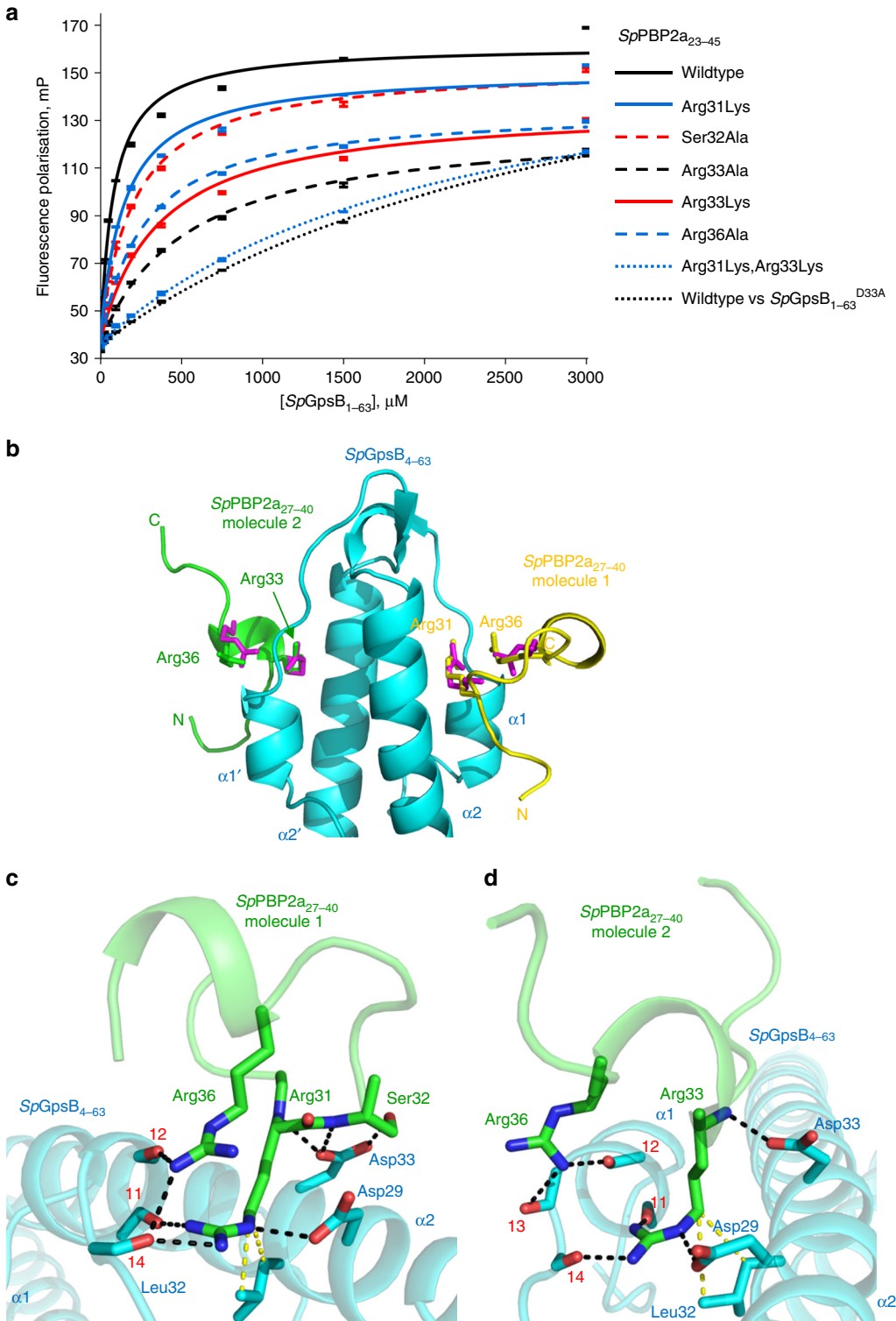

**Fig. 4** The *Sp*PBP2a minidomain is not α-helical but still interacts with *Sp*GpsB through conserved arginines. **a** Arginine residues of *Sp*PBP2a play a key role in binding to *Sp*GpsB. Unless otherwise indicated, the fluorescence polarisation binding curves represent the interaction of TAMRA-labelled *Sp*PBP2a$_{23-45}$ peptides with wildtype *Sp*GpsB$_{1-63}$. The relevant dissociation constants are listed in Supplementary Table 1. **b** The structure of the *Sp*GpsB$_{4-63}$:*Sp*PBP2a$_{27-40}$ complex reveals the critical role of *Sp*PBP2a arginines for the interaction with *Sp*GpsB. In this cartoon, *Sp*GpsB$_{4-63}$ is coloured cyan, and the *Sp*PBP2a$_{27-40}$ peptide is coloured yellow (molecule 1) and green (molecule 2). The sidechains of Arg8 and Arg11 from the *Bs*GpsB$_{5-64}$:*Bs*PBP1$_{1-17}$ complex are shown as red sticks after a global superimposition of equivalent GpsB atoms. In molecule 1, *Sp*PBP2a$^{Arg31}$ and *Sp*PBP2a$^{Arg36}$ superimpose with *Bs*GpsB$_{5-64}$$^{Arg8}$ and *Bs*GpsB$_{5-64}$$^{Arg11}$ whereas molecule 2 accommodates *Sp*PBP2a$^{Arg33}$ and *Sp*PBP2a$^{Arg36}$. **c, d** Close-up view of the interactions of *Sp*PBP2a from molecule 1 (**c**) and 2 (**d**) with *Sp*GpsB$_{4-63}$. Key interfacial sidechains and backbone atoms are represented in stick format; *Sp*GpsB$_{4-63}$ is coloured cyan and *Sp*PBP2A$_{27-40}$ is coloured green. The van der Waals' interactions between *Sp*GpsB$^{Leu32}$ and *Sp*PBP1$^{Arg31}$ (molecule 1) and *Sp*PBP1$^{Arg33}$ (molecule 2) are in yellow. The carbonyl oxygens of *Sp*GpsB$^{Ile11}$, *Sp*GpsB$^{Phe12}$, *Sp*GpsB$^{Glu13}$ and *Sp*GpsB$^{Gln14}$ are denoted by respective red numerals

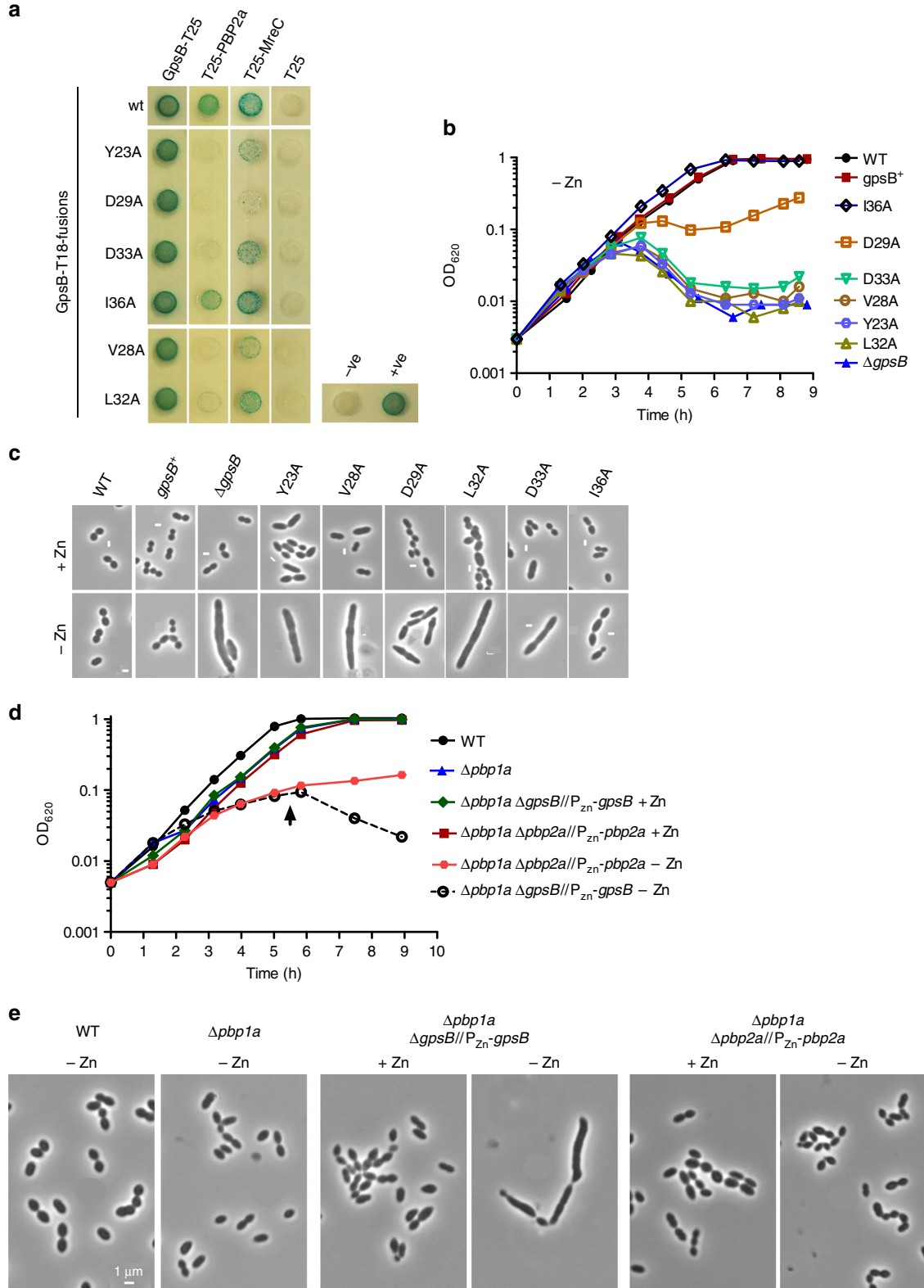

of self-interactions (Supplementary Figure 4C) and were expressed at wild-type levels (Supplementary Figure 4D). However, no obvious phenotype was observed in *S. pneumoniae* strains carrying the corresponding *Sp*PBP2a$^{Arg31Ala}$, *Sp*PBP2a$^{Arg33Ala}$ or *Sp*PBP2a$^{Arg31Ala,Ser32Ala,Arg36Ala}$ alleles, even when *pbp1a* was deleted to decouple the effects of mutations in *Sp*PBP2a from *Sp*PBP1a activity (Supplementary Table 2). Nevertheless,

*Sp*PBP2a mutants in which amino acids 32–37 or 27–38 or 26–45 were deleted in a Δ*pbp1a* background showed progressively reduced growth rates in the three deletion strains and pronounced morphological defects in the two strains with larger deletions (Supplementary Figure 5A, 5B), despite wild-type levels of protein expression at the respective expected molecular masses (Supplementary Figure 5C). At least eight residues, including two

**Fig. 5** Depletion of SpPBP2a does not phenocopy depletion of SpGpsB. **a** Mutations in SpGpsB differentially affect interactions with SpPBP2a and SpMreC. BACTH analysis of the interactions of SpGpsB-T18 variants with wild-type SpGpsB, SpPBP2a and SpMreC. pKT25/pUT18C and pKT25-zip/pUT18C-zip plasmid pairs were used as negative (−ve) and positive (+ve) controls, respectively. The agar plates were photographed after 40h incubation at 30 °C. **b, c** SpGpsB variants that have lost SpPBP2a binding have a gpsB null growth and morphology phenotype. Representative growth curve of S. pneumoniae strains with ectopic expression of gpsB⁺ under a Zn²⁺-dependent promoter. GpsB variants of Y23A, V28A, L32A and D33A showed a gpsB null growth phenotype (**b**) and elongated cell morphology (**c**) on gpsB depletion. The D29A variant showed an intermediate growth phenotype, which was also obtained with an independent isogenic isolate and with a gpsB^D29A-FLAG labelled strain. The I36A strain has a reduced elongation phenotype. All phase-contrast micrographs are at the same magnification (scale bar = 1 μm). **d, e** SpPBP2a depletion does not phenocopy SpGpsB. Representative growth curves (**d**) and phase-contrast micrographs (**e**) of parent IU1824 (WT, D39 Δcps rpsL1), IU13444 (Δpbp1a), IU14381 (Δpbp2a//ΔbgaA::P_Zn-pbp2a⁺ Δpbp1a) and IU14383 (ΔgpsB//ΔbgaA::P_Zn-gpsB⁺ Δpbp1a). Similar to the depletion of SpGpsB in S. pneumoniae pbp1a⁺ strains (see panel **b**), depletion of SpGpsB in IU14383 leads to extremely elongated cells, a growth cessation and lysis phenotype. By contrast, depletion of SpPBP2a in the Δpbp1a background (right hand panels) leads to small but mostly ovococcal cells that do not lyse during the time course examined. All phase-contrast micrographs were taken at OD₆₂₀ ≈ 0.15 or at the time point marked by arrows in **a** for IU14381 and IU14383 under zinc depletion and are at the same magnification (scale bar = 1 μm)

arginines and one lysine, were retained in all the deletion constructs before the predicted start of the SpPBP2a TM helix (residue 54), providing the necessary charge to satisfy the inside-positive rule[43]. Consistent with the pbp2a deletion mutant phenotypes, BACTH results for the correspondent truncated SpPBP2a variants showed reduced interactions with SpGpsB in comparison to the wildtype but not with SpMreC (Supplementary Figure 5D, 5E).

Together, these results support a critical role of the (S/R)RS(R/G)(K/S)xR motif between SpPBP2a residues 30 and 36 for mediating protein–protein interactions with SpGpsB and the importance of its reduction or loss in S. pneumoniae. However, the observation that all three Δpbp1a pbp2a deletion mutants were viable and both growth and morphology phenotypes were different between S. pneumoniae Δpbp1a strains depleted for gpsB (Fig. 5d) and Δpbp1a strains depleted for pbp2a (Fig. 5e) implies that SpPBP2a binding is also just one function for SpGpsB. It is possible that the SpPBP2a:GpsB interaction is only part of the S. pneumoniae ΔgpsB phenotype and this interaction per se is not essential, unless the interaction of GpsB with an additional partner is also lost. Alternatively, since deleting the GpsB binding motif of SpPBP2a did not abolish the interaction with GpsB completely, the deletion might reveal, or even generate, an unpredicted GpsB binding site, perhaps in the juxta-membrane region untouched by the deletions.

Since the pbp2a mutant did not phenocopy gpsB depletion, and given the complex network of GpsB interactions in B. subtilis and L. monocytogenes, we also sought to extend the pneumococcal GpsB interactome starting from those proteins that were reported to form a complex with SpGpsB by co-immunoprecipitation (co-IP)[24]. Besides the SpGpsB interactions detected with SpPBP2a and SpMreC by BACTH, we also detected an interaction between SpGpsB and the essential class B SpPBP2x[44] (Supplementary Figure 6A). SpPBP2x, which contains a cytoplasmic domain abundant with arginines, also interacted with SpGpsB in FP experiments (Supplementary Figure 6B). However, a SpGpsB:SpPBP2x interaction was not detected by co-IP either using SpGpsB-L-FLAG[3] as the bait and an anti-HA antibody to detect HA-tagged SpPBP2x[24] or an anti-SpPBP2x for native SpPBP2x immnodetection, or SpPBP2x-FLAG[3] as bait and an anti-SpGpsB antibody for immunodetection of SpGpsB (Supplementary Figure 6C). These results suggest that SpGpsB:SpPBP2x complexes may be insufficiently abundant, or present only transiently, to be detected by co-IP in pneumococcal cells. SpGpsB also interacted in BACTH with the class A SpPBP1a (Supplementary Figure 6A), which contains a LRLIKY motif in a putative helical region in its small cytoplasmic domain of 18 amino acids. On the other hand, no direct interaction was detected between SpGpsB and SpPBP2b either by BACTH (Supplementary Figure 6A) or by FP (Supplementary Figure 6B), consistent with the lack of any

candidate GpsB-binding sequence motif in SpPBP2b. The SpPBP2b:SpGpsB interaction detected previously by co-IP[24] (Supplementary Figure 6C) is therefore likely to be indirect, presumably involving a bridging protein. The SpGpsB^Asp33Ala allele that resulted in a complete loss of binding to SpPBP2a by FP (Supplementary Table 1) also abrogated interactions in BACTH with a markedly greater impact on interactions with SpPBP2a compared to SpMreC (Supplementary Figure 6A). SpGpsB^Tyr23Ala and SpGpsB^Asp29Ala mutations had a greater impact on SpMreC interactions than SpGpsB^Asp33Ala; this pattern mirrors the impact of the equivalent mutations (LmGpsB^Tyr27Ala, LmGpsB^Asp33Ala and LmGpsB^Asp37Ala) on MreC interactions in L. monocytogenes (Supplementary Figure 3D) and thus substantiates the evidence that different surfaces are used by GpsB to bind a specific class A PBP and MreC. Whilst the interactions of SpGpsB with SpEzrA and SpStkP were confirmed by BACTH, and observed with the reciprocal hybrid pairs, they were unaffected by any of the SpGpsB alleles mentioned above (Supplementary Figure 6A). The interactions of GpsB in the pneumococcus, as detected by BACTH, FP and co-IP, are summarised in Supplementary Figure 6D.

Taken as a whole, our data on three important bacterial systems agree that GpsB is an adaptor protein[29–31] that connects a major class A PG synthase with other cell wall and cell cycle proteins, and to cell shape determinants such as MreC. The identity and mode of interaction of the GpsB-binding partners varies from species to species and may reflect the different physiologies of each bacterium and their modes of growth and division.

## Discussion

Bacterial cell growth and division necessitates tight co-ordination between the replication and segregation of the chromosome, the fission of the cell membrane and the remodelling of the PG. Consequently proteins and their complexes with major functions on either side of the membrane must co-ordinate their activities. One potential mechanism involves the interactions of major PG synthases with their intracellular regulators. Herein we present the first structures of the cell cycle adaptor, GpsB, in complex with the cytoplasmic mini-domains of PG synthases from three different bacteria, the rod-shaped B. subtilis and L. monocytogenes, in which gpsB is conditionally essential[11,12,19] and the ovococcal S. pneumoniae in which gpsB is essential[22–24]. In common with mammalian adaptors GGA[31] and 14-3-3[45] proteins, the primary binding surface of GpsB is restricted to a conserved groove between α-helices The cytoplasmic mini-domains of the three PG synthases in the three organisms have little in common except that each utilises a conserved arginine in their respective sequences to interact with the cognate GpsB. The PG synthase arginine finger pokes into a negatively charged cavity

situated between α-helices 1 and 2 of GpsB and is fixed in the same orientation in all structures, just as the phosphoryl group defines the binding orientation of peptides to 14-3-3[45]. The arginine complements the cavity best when the mainchain amide protons of it and its downstream residue are accessible to form hydrogen bonds with $Bs$GpsB$^{Asp35}$, $Lm$GspB$^{Asp37}$ or $Sp$GspB$^{Asp33}$. This scenario can occur when the arginine is either at the start of an α-helix, such as $Bs$PBP1$^{Arg8}$, or at the $i + 1$ position in a type I β turn, such as $Sp$PBP2a$^{Arg31}$, which explains why free L-arginine does not displace pre-bound PBP peptides from GpsB even when present at 100-fold molar excess (Supplementary Figure 2E). Similarly, contact to the backbone amide at the $i + 2$ position in 14-3-3 ligands is essential for binding[45]. Despite a lack of strong sequence and structural homology in the PG synthase cytoplasmic mini-domains, their binding is dependent upon an identical subset of GpsB residues including $Bs$GpsB$^{Tyr25}$, $Bs$GpsB$^{Asp31}$, $Bs$GpsB$^{Asp35}$ (Fig. 1c) and their structural equivalents $Lm$GspB$^{Tyr27}$, $Lm$GspB$^{Asp33}$, $Lm$GspB$^{Asp37}$ (Fig. 2a), and $Sp$GspB$^{Tyr23}$, $Sp$GspB$^{Asp29}$ and $Sp$GspB$^{Asp33}$ (Fig. 4c, d, Supplementary Figure 1A). Mutations at these positions in GpsB from $S. pneumoniae$ and $L. monocytgenes$ are lethal or have marked growth defects (Fig. 5b, c)[19] and do not interact with their cognate peptidoglycan synthase by FP (Supplementary Table 1)[19] or by BACTH (Fig. 5a)[19]. A phenotype analysis of equivalent mutations in $B. subtilis$ to assess their importance is not straightforward to determine for two reasons. First, a phenotype for the $gpsB$ null mutant is only observed synthetically, when $gpsB$ is mutated alongside either $ftsA$[12] or $ezrA$[11]. Second, a $B. subtilis$ mutant lacking all class A PBPs is viable[46], precluding an unequivocal assessment of the precise importance of the $Bs$GpsB:$Bs$PBP1 interaction. Nonetheless, the convergence of all our experimental data on three bacterial systems suggests strongly that the tyrosine and aspartate dyad also have important roles in $B. subtilis$. Furthermore, these amino acids are also conserved in the DivIVA/Wag31/antigen 84 actinobacterial homologues of GpsB, suggesting a role for them in recruiting cell wall synthesis enzymes to the hyphal tip and future branch sites[13,14], regions that require nascent PG synthesis in filamentous bacteria.

We originally set out to establish the common rules by which GpsB interacts with major PG synthases. Other than the arginine finger mode of GpsB recognition, we discovered how GpsB interacts with at least one class A PBP in each species; that $Lm$GpsB interacts with both the cell shape determinant, MreC[47], and a regulator of Z-ring dynamics, EzrA[10], while in $S. pneumoniae$ we have confirmed the $Sp$GpsB:MreC interaction and newly identified interactions with the class A $Sp$PBP1a and with the essential class B $Sp$PBP2x[44]. These new data complement what was previously known about these interactions in $B. subtilis$[11] and $S. pneumoniae$[23,24]. How GpsB can interact with such disparate targets remains unknown but $Bs$GpsB$^{Asp31}$, $Lm$GpsB$^{Asp33}$ and $Sp$GpsB$^{Asp29}$ are important for interactions with PBPs and other proteins, including MreC, while $Lm$GpsB$^{Asp37}$ and $Sp$GpsB$^{Asp33}$ only interact with PBPs. There must be at least one other surface that is used by GpsB to form complexes with other proteins in its function as an adaptor.

The GpsB:PBP interaction interface notably requires no more than three sidechains from any PBP to complex with GpsB. Protein:peptide contacts involving less well-conserved exosites that flank a small core, conserved peptide motif can contribute significantly to the affinity of protein:peptide interactions[48,49], and the contribution of exosites to affinity, such as the juxtamembrane region, may explain why point mutations in $Lm$PBPA1 and $Sp$PBP2a have a significant impact using peptide fragments in vitro, but have reduced impact in bacterial cells. For instance, the $Lm$PBPA1$^{Arg8Ala}$ mutation had negligible effect (- Fig. 2c, Supplementary Figure 2D) yet it reduced binding by > 15-

fold (Supplementary Table 1). Similarly $Sp$PBP2a$^{Arg31Lys,Arg33Lys}$ had a > 25-fold impact on binding (Supplementary Table 1) yet growth phenotypes were not evident (Supplementary Table 2) until significant stretches of the cytoplasmic mini-domain were deleted (Supplementary Figure 5B).

We also found some differences in GpsB interactions between the species that may be related to GpsB species-specific function. In $B. subtilis$, we discovered a critical motif, SRxxR(R/K), found in close proximity to the membrane that could be used to predict novel GpsB partners. We used this information to identify an interaction network involving cell envelope binding and modifying proteins that most likely is underpinned by the GpsB hexamer. An RSxxxR motif was identified in class A PBPs from most streptococci and sequence features that could dictate GpsB-binding can be found in class A PBPs in other Gram-positive organisms such as the lactococci, $Leuconostocaceae$ and enterococci, including the ESKAPE pathogen $E. faecium$ (Supplementary Figure 7). However, sequence-based searches alone will not identify complete GpsB interactomes because the local structure of the sequence and its proximity to the membrane are also key parameters of GpsB binding. $Sp$PBP2a contains a partially conserved sequence RSxxxR (Supplementary Figure 4A) that resembles the $Bs$PBP1 signature motif and the $Sp$PBP2a mini-domain interacts with each subunit of the $Sp$GpsB dimer in a different way (Fig. 4b–d). We have observed here that mutation or deletion of the cytoplasmic mini-domain of $Sp$PBP2a does not phenocopy deletion or depletion of $gpsB$ (Fig. 5c, e). Similarly, a Δ$pbpA1$ strain did not phenocopy $gpsB$ deletion in $L. monocytogenes$[19]. Taken together these results imply that there must be at least one other critical GpsB interaction partner, beyond respective class A PG synthases, that dictates its conditional essentiality in $L. monocytogenes$ and $B. subtilis$, and essentiality under normal growth conditions in $S. pneumoniae$.

In every Firmicute (and Actinobacterium) tested thus far, apart from $S. aureus$[26], GpsB (or the homologous DivIVA/Wag31/antigen 84) acts as an adaptor to co-ordinate PG synthase activity with other processes depending on the physiology of the cell. GpsB hexamerisation can bridge the interaction of multiple binding partners, a function GpsB shares with 14-3-3 proteins that can form ternary complexes with BCR and Raf-1 by 14-3-3 dimerisation[50]. In bacilli, $Bs$GpsB plays a role in shuttling between the side wall during elongation and the septum during division[11] and, given that $Bs$PBP4b is regulated by σ factors E[35] and F[51], complexes of $Bs$PBP4b and $Bs$YrrS, bridged by $Bs$GpsB (Fig. 3c, d), presumably play a role in the asymmetric cell division characteristic of endospore-forming bacilli. In listeria, which is closely related to bacilli and shares with them a rod-like morphology, GpsB appears to connect several PBPs with proteins with known roles in cytokinesis, including Z-ring polymerisation modulators (ZapA, EzrA, SepF), late division proteins (DivIB, DivIC) and the elongasome (MreC, MreBH) (Supplementary Figure 3C,D), all of which except SepF and MreBH have also been tested in $B. subtilis$ and found not to interact with $Bs$GpsB[11]. By contrast, $S. aureus$ GpsB appears to modulate Z-ring assembly[26], however, no interaction between FtsZ and GpsB has been identified in $B. subtilis$[11], $S. pneumoniae$[23,24] or $L. monocytogenes$ (Supplementary Figure 3C).

The pneumococci have an ovoid cell shape and lack key components such as the MinCD system for cell division site selection[52], and MreB-like proteins required for side wall synthesis[53]. Presumably $Sp$GpsB interacts with one or more pneumococcal-specific proteins, the loss of which may be related to the lethal phenotype. Furthermore, $Sp$GpsB affects both StkP autophosphorylation[23,24] and the StkP-catalysed phosphorylation of $Sp$DivIVA[23,24], $Sp$MapZ/LocZ[24,54], $Sp$Jag/EloR/KhpB[55–57] and $Sp$MacP[58]. It is not yet clear how the complexes formed by these

proteins are affected by their phosphorylation, except that $Sp$PBP2a activity is dependent upon phosphorylated $Sp$MacP[58], at least in the presence of functional $Sp$StkP, or what the impact is of potential cross-talk to two-component signalling systems[59].

Finally, the different phenotypic outcomes associated with $gpsB$ deletion or depletion in the three systems studied herein may reflect the presence of redundant systems in the large genome (4.2 Mbp) of the bacilli, partial redundancy in listeria (2.9 Mbp), and a relative absence of redundancy in the stripped-down genome (2.1 Mbp) of the pneumococci. The relative affinities and cellular concentrations of GpsB partners probably dictate which protein (s) is bound by GpsB at any point in a cell cycle-dependent manner; simultaneous interactions with multiple target proteins is likely to lead to an increase in avidity of GpsB[19] as commonly found in antibody:antigen interactions. However, the intricate networks involving GpsB will only be uncovered by validating the full GpsB interactome.

## Methods

**Bacterial strains and growth conditions.** Supplementary Table 3 lists the bacterial strains used in this study. $L.$ $monocytogenes$ EGD-e-derived strains were routinely cultivated in brain heart infusion (BHI) broth or on BHI agar plates at 37 °C or 42 °C, where indicated. If required, erythromycin (5 µg$^{-1}$mL$^{-1}$) and X-Gal (40–100 µg$^{-1}$mL$^{-1}$) were added. All $L.$ $monocytogenes$ growth experiments were repeated three times and average values and standard deviations are shown. All $S.$ $pneumoniae$ strains were derived from unencapsulated serotype 2 D39 strains IU1824 (D39 $\Delta cps$ $rpsL1$) or IU1945 (D39 $\Delta cps$)[60]. $S.$ $pneumoniae$ strains were grown on Petri dishes containing the appropriate antibiotic, modified trypticase soy agar II (Becton-Dickinson) and 5% (vol per vol) defibrinated sheep blood (TSAII-BA), and the plates were incubated at 37 °C in 5% $CO_2$. Bacteria were cultured statically in BHI (Becton-Dickinson) broth at 37 °C in an atmosphere of 5% $CO_2$, and growth was monitored by $OD_{620}$. $Escherichia$ $coli$ TOP10 and DH5α (lab stocks) were used as standard hosts for all cloning procedures[61], and BL21(DE3) strains (lab stocks) were used for recombinant protein production.

**General methods, manipulation of DNA and oligonucleotide primers.** $E.$ $coli$ transformation and isolation of plasmid DNA (the plasmids used are listed in Supplementary Table 4) was performed according to standard protocols[61]. $L.$ $monocytogenes$ strains were transformed by electroporation. Enzymatic modification of plasmid DNA was carried out as described by the instructions given by the manufacturers. Quikchange mutagenesis was employed for restriction-free modification of plasmids[62]. DNA sequences of oligonucleotide primers are listed in Supplementary Tables 5–7. All deletions and insertions were confirmed by PCR, and all constructs and clones were verified by Sanger DNA sequencing.

**Construction of $L.$ $monocytogenes$ mutant strains.** Plasmid pSH497 was first constructed to facilitate mutagenesis of the chromosomal copy of $pbpA1$. A DNA fragment comprising $recU$ and the first 2063 bp of $pbpA1$ was amplified from chromosomal DNA using primer pair SHW773/SHW774 and cloned into pMAD using $BamHI/NcoI$. An unwanted mutation in front of $recU$ resulted in a correction of this plasmid by Quikchange with primer pair SHW777/SHW778. The corrected pSH497 was used as template in further Quikchange reactions to introduce Thr7Ala (SHW744/SHW745, pSH504), Arg8Ala (SHW746/SHW747, pSH505), Tyr11Ala (SHW755/SHW756, pSH506), Gln10Pro (SHW787/SHW788, pSH509) and Arg12Ala mutations (SHW748/SHW749, pSH507) into $pbpA1$, or to remove $pbpA1$ nucleotides 4-1250 (SHW775/SHW776, pSH503). pSH508 ($pbpA1^{\mathrm{Arg8Ala/}}$$^{\mathrm{Arg12Ala}}$) was obtained by Quikchange with primers SHW750/SHW751 and plasmid pSH505 as the template. An N-terminal fragment of $pbpA1$ was excised from the $L.$ $monocytogenes$ chromosome using plasmid pSH503 and the insertion/excision protocol for construction of clean deletions, resulting in strain LMS211. The mutated $pbpA1$ alleles present on pSH504-pSH509 were then reintroduced into LMS211 following the same protocol.

**$L.$ $monocytogenes$ fosfomycin susceptibility assays.** Fosfomycin susceptibility was recorded using filter discs (Ø 6 mm) soaked with 10 µL of a 10 mg$^{-1}$mL$^{-1}$ fosfomycin solution. $L.$ $monocytogenes$ colonies, grown on BHI agar plates, were resuspended in BHI broth and used to swab-inoculate BHI agar plates. Filter discs soaked with fosfomycin were placed on top of the agar surface and the plates were incubated at 37 °C overnight. Diameter of growth inhibition zones was measured and corrected for the filter disc diameter. All experiments were performed three times and average values and standard deviations were calculated. Significance was determined using the $t$-test and differences were considered to be significant when $P < 0.01$.

**$L.$ $monocytogenes$ and $B.$ $subtilis$ BACTH assay.** The BACTH system[63] was used for analysis of protein:protein interactions. First, $Lm$GpsB was screened against $Lm$PBPA1 ($lmo1892$) variants to identify residues in $Lm$PBPA1 that are important for binding $Lm$GpsB. To facilitate screening, we used a T25 fragment of $Bordetella$ $pertussis$ adenylate cyclase that had been fused to the N-terminal 91 amino acids of $Lm$PBPA1 to create a T25-$Lm$PBPA1 fusion containing the N-terminal cytoplasmic mini-domain and the transmembrane helix, but lacking lacking the extracellular glycosyltransferase and transpeptidase domains (T25-PBPA1ΔGT-TP) in plasmid pSH437[21]. $Lm$PBPA1 residues that corresponded to the $Bs$GpsB binding motif of $Bs$PBP1 (Thr7, Arg8, Tyr11, Arg12) and additional positively charged amino acids (Lys14, Lys20, Lys21, Lys22, Lys25, Arg26, Lys28 and Arg29) in T25-PBP A1ΔGT-TP were replaced with alanine prior to screening against GpsB-T18. The pSH437 plasmid was mutated by Quikchange mutagenesis using the primer pairs SHW744/SHW754 (T7A, pSH485), SHW746/SHW747 (R8A, pSH486), SHW755/SHW756 (Y11A, pSH487), SHW748/SHW749 (R12A, pSH488), SHW757/SHW758 (K14A, pSH489), SHW759/SHW760 (K20A, pSH490), SHW761/SHW762 (K21A, pSH491), SHW763/SHW764 (K22A, pSH492), SHW765/SHW766 (K25A, pSH493), SHW767/SHW768 (R26A, pSH494), SHW769/SHW770 (K28A, pSH495) and SHW771/SHW772 (R29A, pSH496).

pJR164 and pJR169 were constructed by cloning $sepF$ (primers JR334/JR335 and $XbaI/KpnI$ digestion) into pKT25 and p25-N, respectively. pJR165 and pJR170 were generated by cloning $ezrA$ into pKT25 and p25-N, respectively, using primers JR339/JR340 and $XbaI/KpnI$ for digestion. pJR172 and pJR174 were obtained by cloning $zapA$ into pKT25 and p25-N, respectively, using primers JR337/JR338 and $PstI/KpnI$. pJR213 was constructed by amplification of $divIB$ with primers JR353/JR354 and cloning into pKT25 using $XbaI/KpnI$. pJR214 was generated by PCR amplification of $divIC$ with primers JR355/JR356 and cloning into pKT25 using $XbaI/KpnI$. $mreBH$ was amplified by PCR using oligonucleotides JR368/JR369 and cloned into pKT25 and p25-N using $XbaI/KpnI$, resulting in pJR233 and pJR236, respectively. pJR242 and pJR243 were obtained by cloning $mreC$ into pKT25 and p25-N, respectively, using primers JR364/JR365 and $PstI/KpnI$ digestion. $pbpA2$ was cloned into pKT25 using oligonucleotides SHW153/SHW154 and $XbaI/KpnI$ to create pJR250. $pbpB1$ was amplified with primers SHW157/SHW158 and cloned with $SmaI/EcoRI$ into pKT25, yielding pSH236. $pbpB2$ was amplified with primers SHW155/SHW156 and cloned into pKT25 after $PstI/BamHI$ digestion, yielding plasmid pSH235. $pbpB3$ was amplified using primers SHW159/160 and cloned by $PstI/BamHI$ digestion into pKT25, resulting in pSH237. Finally, pSH484 was obtained by removing the N-terminal domain from $gpsB$ present on plasmid pSH226 in a PCR with the primer pair SHW752/SHW753.

Plasmids encoding the respective genes fused to the N- or C-termini of the T18- or the T25-fragment of the $Bordetella$ $pertussis$ adenylate cyclase were co-transformed into $E.$ $coli$ BTH101. Transformants were selected on LB agar plates containing ampicillin (100 µg$^{-1}$mL$^{-1}$), kanamycin (50 µg$^{-1}$mL$^{-1}$), X-Gal (40 µg$^{-1}$mL$^{-1}$) and IPTG (0.1 mM) and photographs were taken after 24 and 48h growth at 30 °C. The BACTH experiments in Fig. 2b and Supplementary Figure 3C, D were repeated at least twice.

The same procedures outlined above were followed for the BACTH using $B.$ $subtilis$ GpsB except that co-transformants were grown on nutrient agar plates[11], not LB, before imaging. These experiments were replicated reproducibly at least three times and a representative image is provided in Fig. 3c. Reciprocal tests for some interactions do not show the same results presumably because of the different expression levels for the individual fusion constructs as pKT25 is a low copy number plasmid whereas pUT18C is a high copy number plasmid.

**$S.$ $pneumoniae$ strain construction.** Mutant strains containing antibiotic resistance markers were constructed by transformation of CSP-1 induced competent pneumococcal cells with linear DNA amplicons synthesised by overlapping fusion PCR[64]. Mutant constructs were confirmed by PCR and DNA sequencing of chromosomal regions corresponding to the amplicon region used for transformation.

Ectopic expression of wildtype $gpsB$ with a $Zn^{2+}$-inducible promoter (IU11286) was achieved by insertion into $bgaA$ to create a $\Delta bgaA::tet$-$P_{Zn}$-$gpsB^+$ construct using PCR-fused fragments. To investigate whether GpsB with allele changes of Y23A, V28A, D29A L32A D33A or I36A were functional, and given that $gpsB$ is essential for the growth of $S.$ $pneumoniae$ D39, a merodiploid D39 $\Delta cps$ $\Delta gpsB$<>$aad9$//$\Delta bgaA::P_{Zn}$-$gpsB^+$ strain (IU11388) was constructed by transforming $\Delta gpsB$<>$aad9$ into IU11286 in the presence of 0.5 mM $ZnCl_2$ + 0.05 mM $MnSO_4$, which is required in the media to counteract the $Zn^{2+}$ toxicity[57,64]. For selection of $gpsB$ alleles, a $P_c$-$erm$ cassette was fused to the 3′ end of $gpsB$ with the desired mutations in the native chromosomal locus. The $gpsB$ allele change-$P_c$-$erm$ amplicon was transformed into IU11388 in the presence of erythromycin and 0.5 mM $ZnCl_2$ + 0.05 mM $MnSO_4$. The $P_c$-$erm$ cassette inserted at the 3′ end of $gpsB^+$ imparted no discernible growth phenotype compared to a wild-type strain (Fig. 5b). The growth of merodiploid strain IU11388, and strains containing $gpsB$ with allele changes complemented with ectopic expression of WT $gpsB$ (IU12361, IU12363, IU12440, IU12612, IU12615, IU13121), or $gpsB$ with allele changes linked to a FLAG tag complemented with ectopic expression of WT $gpsB$ (IU13141, IU13364, IU13366, IU13368, IU13370, IU13372 and IU13374) were performed in the presence of 0.5 mM $ZnCl_2$ + 0.05 mM $MnSO_4$. Transformation of $\Delta pbp1a::$

$P_c$-erm into IU11388 ($\Delta gpsB<>aad9//\Delta bgaA::tet$-$P_{Zn}$-$gpsB^+$) and IU14365 ($\Delta pbp2a$ markerless//$\Delta bgaA::kan$-$P_{Zn}$-$pbp2a^+$) was performed in the presence of 0.4 mM $ZnCl_2$ + 0.04 mM $MnSO_4$. Strains containing markerless $pbp2a$ alleles in the native chromosomal locus were constructed using Janus cassette allele replacement[65] via an $rpsL1$ strain containing a $\Delta pbp2a::P_c$-[$kan$-$rpsL^+$] construct (IU7853).

**Transformation of a $\Delta pbp1a$ amplicon into *S. pneumoniae* strains.** The function of the mutated $pbp2a$ alleles was evaluated by transformations of 30 ng of a $\Delta pbp1a::P_c$-erm amplicon, obtained from strain E177 and containing ≈1 kb of flanking chromosomal DNA, into strains harbouring $pbp2a$ mutations. Transformants were visualised after 24 h incubation at 37 °C in 5% $CO_2$. The numbers of colonies were normalised to 1 mL of transformation mixture.

**Growth and microscopy of *S. pneumoniae* strains.** Strains IU1945 (wild-type parent) and IU11286 ($gpsB^+//\Delta bgaA::P_{Zn}$-$gpsB^+$) were inoculated from frozen glycerol stocks into BHI broth, serially diluted, and incubated for <13 h statically at 37 °C in an atmosphere of 5% $CO_2$. For IU11388 ($\Delta gpsB//\Delta bgaA::P_{Zn}$-$gpsB^+$) and all $gpsB$ allele exchange strains complemented with $bgaA::P_{Zn}$-$gpsB^+$, 0.5 mM $ZnCl_2$ + 0.05 mM $MnSO_4$ (+Zn) was added to BHI broth cultures to enable the ectopic expression of $gpsB$[64]. The next day, cultures at $OD_{620} \approx 0.1$–0.4 were diluted to $OD_{620} \approx 0.003$ in BHI broth lacking (−Zn) or containing (+Zn) 0.5 mM $ZnCl_2$ + 0.05 mM $MnSO_4$ and cultured under the same conditions. Growth was monitored turbidimetrically every 45 min to 1 h with a Genesys 2 spectrophotometer (Thermo Scientific). At 3.75 h after dilution into BHI broth with or without Zn/Mn supplement, cell pellets were obtained from 1 mL of culture, and resuspended in 100 µL of 4% paraformaldehyde. Cells were fixed for 15 min at RT and the tubes with fixed cells were left on ice until microscopy. For microscopic analyses, samples (1.5 µL) were taken and examined using a Nikon E-400 epifluorescence phase-contrast microscope. Growth and microscopy of strains containing $pbp2a$ alleles were performed as above, but with no Zn/Mn addition to the BHI broth. Microscopy was performed at $OD_{620} \approx 0.15$.

Growth and microscopy experiments of IU14381 ($\Delta pbp2a//P_{Zn}$-$pbp2a^+$ $\Delta pbp1a$) and IU14383 ($\Delta gpsB//P_{Zn}$-$gpsB^+$ $\Delta pbp1a$) were performed as for IU11388 ($\Delta gpsB//P_{Zn}$-$gpsB^+$) except that overnight growths were carried out in BHI broth containing 0.2 mM $ZnCl_2$ + 0.02 mM $MnSO_4$ for IU14381, and 0.4 mM $ZnCl_2$ + 0.04 mM $MnSO_4$ for IU14383. Cultures were diluted into the same media as the overnight media for +Zn condition and into BHI broth without Zn/Mn for the −Zn condition. All growth measurements were reproducible and were repeated at least three times. The mean doubling time and the standard error of the mean ( ± ) are reported in Supplementary Figure 5B.

**Western blotting and immunodetection of *S. pneumoniae* strains.** To confirm the expression of $gpsB$ with Y23A, V28A, D29A, L32A, D33A or I36A allele exchanges, western blot analyses were performed of strains constructed with $gpsB$-FLAG or $gpsB$ variant-FLAG at the native site complemented by ectopic Zn-dependent expression of WT $gpsB$. 0.5 mM $ZnCl_2$ + 0.05 mM $MnSO_4$ (+Zn) was added to all BHI broth cultures to enable the ectopic expression of $gpsB$, and were diluted to $OD_{620} \approx 0.003$ and cultured under the same conditions to $OD_{620} \approx 0.15$ to 0.2. Under the +Zn condition, all FLAG-tagged $gpsB$ allele exchange strains grew identically to WT $gpsB^+$ strain and to the untagged $gpsB$ allele exchange strains. A parallel set of cultures was performed using BHI broth without addition of $ZnCl_2$ and $MnSO_4$ to confirm that FLAG-tagged $gpsB$ allele strains grew identically to their untagged counterparts. Aliquots of 1.8 mL were centrifuged (5 min, 16,000 × g at 4 °C), and cell pellets were washed once with PBS. The supernatant was discarded and the pellets were frozen on dry ice for 5 min. The pellets were then thawed for 5 min at RT, resuspended in 100 µL of prewarmed 37 °C SEDS lysis buffer (0.1% deoxycholate, 150 mM NaCl, 0.2% SDS, 15 mM EDTA pH 8.0)[24]. Samples were incubated on a 37 °C shaking block at 300 rpm for 15 min, and were vortexed vigorously twice during the 15 min incubation. Total protein concentrations were determined using a Bio-Rad DC™ protein assay kit. 4 µg of total protein was loaded per lane on a 4–15% precast gradient SDS-PAGE gel (Bio-Rad), subjected to electrophoresis and transferred to a nitrocellulose membrane. FLAG-tagged GpsB proteins were detected with an anti-FLAG rabbit polyclonal antibody (Sigma, F7425, 1:1400 dilution) as primary antibodies, and ECL anti-rabbit IgG horseradish peroxidase linked whole antibody as secondary (dilution 1:10,000). Chemiluminescent signals in protein bands were detected with an IVIS imaging system. MreC was detected with anti-MreC antibodies[66] by relabelling the nitrocellulose membrane after detection with anti-FLAG.

Western detection of PBP2a in various PBP2a variant strains was performed essentially as above with exponentially growing cultures in BHI broth. 4 µg of total protein was loaded onto a 10% precast SDS-PAGE gel (Bio-Rad). Electrophoresis was carried out for 1.5 h to allow separation of $Sp$PBP2a from a non-specific band. Anti-$Sp$PBP2a serum[67] was used at a dilution of 1:5000.

The anti-serum against purified, recombinant $Sp$GpsB$_{1-63}$ was characterised by preparing lysates from $gpsB^+$ $divIVA^+$ WT (strain IU1945), $\Delta gpsB$ (IU6442) or $\Delta divIVA$ (IU8496) strains as described above. 4 µg of total protein was loaded per lane on a 4–20% precast gradient SDS-PAGE gel (Bio-Rad), and anti-GpsB serum from rabbit Ab-1432 was used at a dilution of 1:2000.

**Co-immunoprecipitation (Co-IP) of *S. pneumoniae* strains.** Co-IP experiments of *S. pneumoniae* FLAG-tagged strains were performed with the use of anti-FLAG magnetic beads[24]. Lysates were obtained from cultures grown exponentially at 37 °C in an atmosphere of 5% $CO_2$ in 400 mL of BHI to $OD_{620} \approx 0.25$–$0.40$[24]. Cell pellets were washed once with 30 mL of 1X PBS (4 °C) and resuspended in 19.8 mL 1× PBS (4 °C). About 200 µL of 10% (vol per vol) paraformaldehyde solution (EMS) were added for crosslinking to a final concentration of 0.1% (vol per vol). Mixtures were incubated at 37 °C in an air incubator for 1 h. Cross-linking reactions were quenched by the addition of 4 mL 1.0 M glycine followed by incubation at 25 °C for 10 min. Cells were collected by centrifugation (16,500 × g for 5 min at 4 °C). Pellets were washed once with 20 mL cold 1× PBS (4 °C) and resuspended in 2 mL of cold lysis buffer (50 mM Tris-HCl pH 7.4, 150 mM NaCl, 1 mM EDTA, 1% Triton X100 (vol per vol)) with 1 tablet of protease inhibitor (ThermoFisher Scientific, 78429) freshly added per 10 mL of lysis buffer. The suspension was transferred into two lysing matrix B tubes (MP Biomedicals) with 1 mL in each tube. The tubes were shaken ten times in a FastPrep homogenizer (4×, 5 min on ice, 3×, 5 min on ice and 3×) with 6.0 M s⁻¹ for 40 s each at 4 °C. Cell debris and lysing matrix from tubes were removed by centrifuging at 16,000 × g for 5 min at 4 °C. The protein concentration of each sample was determined by Bio-Rad DC™ protein assay (Bio-Rad). About 1 mL of lysate with similar amounts of total protein (5–7 g⁻¹mL⁻¹) was added to tubes with 50 µL of anti-FLAG magnetic beads (Sigma, M8823). The same amount of protein was loaded onto the beads for strains expressing FLAG-tagged proteins and the corresponding control strains lacking FLAG-tagged proteins in each experiment. The tubes were rotated for 2 h at 4 °C. The beads were washed three times with 1 mL of lysis buffer (4 °C) with 10 min incubation at 4 °C each time. FLAG-tagged proteins were eluted from the beads by incubation with 100 µL of FLAG elution solution (150 ng 3X FLAG peptide per µL) (Sigma, F4799) for 30 min at 4 °C. A volume of 100 µL of the elution and of the original lysate added to magnetic beads (input) were separately mixed with 100 µL 2× Laemmli sample buffer (Bio-Rad) containing 5% (vol per vol) β-mercaptoethanol (Sigma) and heated at 95 °C for 1 h to break the cross-links. A volume of 40 µL of each elution sample mixed with 2× sample buffer was loaded on each lane of a 4–15% precast SDS-PAGE gel and subjected to western blotting using affinity-purified $Sp$PBP2x polyclonal antibodies (dilution 1:10,000)[67] or an anti-GpsB antibody as the primary antibodies.

**S. pneumoniae BACTH assay.** The target genes were amplified by the PCR from *S. pneumoniae* D39 chromosomal DNA. PCR fragments for $pbp2a$, $mreC$, $pbp1a$, $pbp2x$ and $pbp2b$ were purified, digested with appropriate restriction enzymes and cloned into the corresponding sites of the pKT25/pUT18C vectors to generate plasmids encoding the corresponding hybrid proteins fused at the C-terminal ends of the T25 and T18 fragments, respectively. Plasmids pKNT25-$gpsB$/pUT18-$gpsB$, pKNT25-$stkP$/pUT18-$stkP$ and pKNT25-$ezrA$/pUT18-$ezrA$ were already constructed[24]. The mutated and truncated $gpsB$ and $pbp2a$ alleles were amplified from their respective DNA templates and the corresponding PCR products cloned into the corresponding sites of the BACTH vectors as described above for the wild-type alleles. *E. coli* DH5α transformants were selected on LB agar plates containing ampicillin (100 µg⁻¹mL⁻¹) or kanamycin (50 µg⁻¹mL⁻¹) and 0.4% glucose to repress leaky expression[63].

BACTH experiments, each pair of plasmids was co-transformed into the *E. coli* cya- strain BTH101 and co-transformation mixtures were spotted onto LB agar plates supplemented with ampicillin (100 mg⁻¹mL⁻¹), kanamycin (50 mg⁻¹mL⁻¹) and X-Gal (40 µg⁻¹mL⁻¹), followed by incubation at 30 °C. Plates were inspected and photographed after 24 and 40 h. Plasmid pairs pKNT25/pUT18 and pKT25-zip/pUT18C-zip were used as negative and positive controls, respectively. All experiments were repeated at least twice.

**Plasmid construction for recombinant protein and peptide work.** (i) GpsB: All mutagenesis was undertaken by the Quikchange protocol; where mutagenesis reactions failed to generate a DNA fragment of the expected size, a modification of the Quikchange protocol with two separate PCR steps was used instead[69]. Glu17Ala, Asp31Ala and Asp35Ala mutations were introduced to the $Bs$GpsB$_{1-68}$ expression construct[19] by Quikchange. The $Bs$GpsB$_{5-64}$ construct was prepared by PCR amplification of $gpsB$ from *B. subtilis* 168 genomic DNA using primers BsGpsB5start and BsGpsB64stop. The PCR product was ligated between the *Nde*I and *Xho*I sites of modified pET15b (M26L DJ-1 TEV site pET15b; gift from Mark Wilson, Addgene) with the nucleotides encoding the thrombin cleavage site replaced by a sequence encoding a TEV cleavage site. The $Sp$GpsB$_{1-63}$ expression construct (pET28$Sp$GpsB$_{1-63}$) was prepared by PCR amplification of $gpsB$ from *S. pneumoniae* R6 genomic DNA with primers SpGpsB5ndeI and SpGpsB3XhoI. This PCR product was ligated between the *Nde*I and *Xho*I sites of pET28a, and a stop codon was introduced by Quikchange mutagenesis in place of the codon for Pro64. The $Sp$GpsB$_{4-63}$ expression construct was generated in two steps; first the ORF encoding residues 1–63 of $Sp$GpsB was amplified with primers SpGpsBM11NcoI and SpGpsBM11XhoI using plasmid pET28$Sp$GpsB$_{1-63}$ as the template. This DNA fragment was then ligated between the *Nco*I and *Xho*I sites of pET11M[70] before the codons for residues 1–3 of $Sp$GpsB were deleted by Quikchange mutagenesis with primers SpGpsB1to3del5 and SpGpsB1to3del3.

(ii) PBP peptides; Ser16Cys, Ser7Ala, Arg8Ala, Ala10Pro, Arg11Ala and Arg28Ala mutations in $Bs$PBP1$_{-32}$ fused to maltose binding protein (MBP)[19] were

introduced by Quikchange mutagenesis. Plasmids expressing $Lm$PBPA1$_{1-20}$ and $Sp$PBP2a$_{23-45}$ peptides fused to MBP were prepared by PCR amplifying the relevant ORFs from $L. monocytogenes$ strain EGD-e and $S. pneumoniae$ R6 genomic DNA with primers $Lm$PBPA1ncoI5, $Lm$PBPA1xhoI3, $Sp$PBP2AncoI5 and $Sp$PBP2AxhoI3, respectively. The subsequent PCR fragments were digested with $Nco$I and $Xho$I for ligating into similarly restricted pMAT11, a modified version of pHAT4[70]. Ser19 and Lys21 in $Lm$PBPA1 were mutated in a single step by Quikchange to cysteine and stop codons, respectively; similarly Quikchange was used to change $Sp$PBP2a Gly43 and Arg46 to cysteine and stop codons, respectively. Further mutations in $Lm$PBPA1 were also made by Quikchange; the Arg8AlaSer16Arg double mutant was prepared in two successive steps with the Arg8Ala mutation made first, followed by Ser16Arg.

The $Sp$PBP2x$_{1-29}$ construct was prepared in several steps. First, nucleotides 1–1165 of $pbp2x$ were PCR amplified in two steps: an initial fragment generated from $S. pneumoniae$ R6 genomic DNA with primers $Sp$PBP2x5/$Sp$PBP2x3st1 was used as a template for further PCR amplification with $Sp$PBP2x5/$Sp$PBP2x3st2 primers; this fragment was then ligated into the $Nco$I and $Xho$I sites of pMAT11. Second, cysteine, serine and stop codons were introduced by Quikchange in place of residues 30–32 with primers $Sp$PBP2xS30CL31SL32STOP5 and $Sp$PBP2xS30CL31SL32STOP3. A second Quikchange step (primers $Sp$PBP2xinsgly5, $Sp$PBP2xinsgly3) introduced extra codons encoding a GSG sequence after the TEV cleavage to improve cleavage of the MBP-PBP2x$_{1-29}$ fusion protein.

(iii) YpbE and YrrS; The construct for expressing YpbE$_{130-240}$ was prepared in two steps; first, the ORF encoding residues 80-240 of YpbE was PCR amplified from genomic DNA and ligated between the $Nde$I and $Xho$I sites of pET28a. Second, the ORF encoding the N-terminal His-tag and residues 80–129 of YpbE was subsequently deleted by PCR amplification of the entire plasmid with primers YpbEtruncatencoi5 and YpbEtruncatencoi3, followed by $Nco$I restriction digestion and religation of the DNA with T4 DNA ligase. The construct for expressing YrrSΔ$_{13-16}$ was prepared by amplifying the $yrrS$ gene from $B. subtilis$ genomic DNA with flanking $Nde$I and $Xho$I sites and then ligating the PCR fragment between the same restriction sites in pET28a. The region coding for residues 13–16 of YrrS was subsequently deleted by Quikchange mutagenesis.

**Recombinant protein purification.** (i) GpsB; $Bs$GpsB$_{1-68}$ proteins used for FP assays were purified by Ni-NTA affinity chromatography followed by proteolytic removal of the His$_6$-tag by thrombin and subsequent size exclusion chromatography[19]. The same protocol was used for $Sp$GpsB$_{1-63}$ except that an ammonium sulphate precipitation step was added after thrombin removal of the His$_6$-tag. Ammonium sulphate was added to the thrombin-cleaved protein to a final concentration of 2.34 M (60% saturation at 0 °C) by adding a stock solution of 100% saturated ammonium sulphate that was prepared by adding solid ammonium sulphate directly to 50 mM Tris.HCl (pH 8), 300 mM NaCl, 10 mM imidazole. After stirring for 30 min at 4 °C the mixture was centrifuged at 19,000 × $g$ for 20 min and solid ammonium sulphate then added to the supernatant to a final concentration of 2.8 M. After stirring at 4 °C for another 30 min the mixture was again centrifuged 19,000 × $g$ for 20 min and further ammonium sulphate added to a final concentration of 3.5 M. After 30 min stirring the mixture was again centrifuged at 19,000 × $g$ and the pellet resuspended in 10 mM Tris.HCl (pH 8), 250 mM NaCl before further purification by gel filtration[19]. $Bs$GpsB$_{5-64}$, $Bs$GpsB$_{5-64}$$^{Lys32Glu}$ and $Sp$GpsB$_{4-63}$ were expressed and purified by a similar protocol except that TEV protease, rather than thrombin, was used to remove the N-terminal His$_6$-tag by overnight cleavage with TEV (1:50 ratio of TEV:GpsB) at 4 °C in a buffer of 50 mM Tris.HCl (pH 8.0), 300 mM NaCl, 250 mM imidazole, 1 mM DTT. The cleaved protein was then dialysed against 20 mM Tris.HCl (pH 8.0), 200 mM NaCl, 10 mM imidazole and manually passed over a 5 mL Ni-NTA superflow cartridge (Qiagen) to remove TEV and uncleaved proteins. The flow through from the column was concentrated and loaded onto a Superdex 75 XK16/60 (GE Health-care) gel filtration column equilibrated in 10 mM HEPES.NaOH (pH 8.0), 100 mM NaCl.

(ii) PBP peptides; The PBP peptides, generated as His-tagged MBP-fusion proteins, were purified by standard Ni-NTA affinity chromatography procedures[19]. The purified MBP-PBP fusions at 5 mg$^{-1}$mL$^{-1}$ were fluorescently labelled in a buffer of 50 mM Tris.HCl (pH 7.0), 300 mM NaCl, 250 mM imidazole by mixing with either TAMRA-maleimide (Santa Cruz Biotechnology) or with fluorescein maleimide (Vector laboratories) to final concentrations of 625 µM and 1 mM from 25 and 40 mM stocks made in DMSO, respectively. After overnight incubation at 4 °C the labelling reaction was exchanged into a buffer of 10 mM Tris.HCl (pH 8.0), 250 mM NaCl, cleaved with TEV protease (1:50) and the labelled peptide separated from the MBP fusion partner by passing through a 3 kDa molecular mass cutoff centrifugal ultrafiltration device[19].

The $Bs$PBP1$_{1-17}$, $Lm$PBPA1$_{1-15}$ and $Sp$PBP2a$_{27-40}$ peptides used for crystallisation and circular dichroism were synthesised chemically (Protein and Peptide Research Ltd, UK and Severn Biotech, UK). The fluorescent-labelled $Bs$YpbE$_{1-21}$ and $Sp$PBP2b$_{1-17}$ peptides used in FP assays were also synthesised chemically (Protein and Peptide Research Ltd); the $Sp$PBP2b$_{1-17}$ peptide encompasses the first seventeen amino acids of $Sp$PBP2b from strain R6, based on the annotated sequence in the NCBI database (accession NP_359110).

(iii) BsPBP1 proteins; Recombinant $Bs$PBP1 proteins were purified from $E. coli$ membrane pellets[19] resuspended in a buffer of 50 mM HEPES.NaOH pH 7.5, 500

mM NaCl, 3 mM MgCl$_2$, 0.3 mM DTT supplemented with Roche complete EDTA-free protease inhibitor cocktail, lysozyme and DNAase. The lysate was clarified by centrifugation at 10,000 × $g$ for 20 min and the supernatant centrifuged further for 1 h at 4 °C at 100,000 × $g$. The membrane pellet was resuspended in lysis buffer supplemented with 15% glycerol, 0.2 mM AEBSF, 0.5 mM DTT and Roche complete EDTA-free protease inhibitor. The resuspended membranes were ultracentrifuged for 1 h at 4 °C at 100,000 × $g$ and the membrane pellet resuspended in resuspension buffer (50 mM HEPES.NaOH pH 7.5, 500 mM NaCl, 3 mM MgCl$_2$, 2% reduced Triton X-100, 15% glycerol, 10 mM β-mercaptoethanol). After 2 h stirring at 4 °C the extract was ultracentrifuged at 100,000 × $g$ for 30 min at 4 °C. The supernatant was supplemented with imidazole to 5 mM then incubated for 2 h at 4 °C with Sigma His select resin. This mixture was then passed through a gravity column and the resin washed with resuspension buffer supplemented with 5 mM imidazole and then $Bs$PBP1 proteins were eluted with resuspension buffer containing 250 mM imidazole. The His$_6$-tag was removed by thrombin treatment, before the digest was purified by Superdex S200 size exclusion chromatography in a buffer of 10 mM HEPES.NaOH pH 7.5, 500 mM NaCl, 3 mM MgCl$_2$, 0.1% reduced Triton X-100, 2% glycerol. Residual uncleaved $Bs$PBP1 proteins were removed by incubation with His select resin before concentraing and snap-freezing in liquid nitrogen.

(iv) YpbE; The YpbE$_{130-240}$ protein and its S132C variant were expressed in BL21(DE3) cells by induction overnight at 17 °C with 0.5 mM IPTG when the culture had reached an optical density at 600 nm of 0.6. The cell pellet from a 2 L culture was resuspended in 70 mL of 50 mM Tris.HCl (pH 8), 10 mM NaCl, supplemented with 1 mL of 25× working concentration of Roche Complete EDTA free protease inhibitor cocktail and lysed by sonication. The lysate was then passed over a XK16 column packed with a 25 mL bed volume of Fast Flow Q-Sepharose and the flow through from the column then mixed with an equal volume of lysis buffer supplemented with saturated ammonium sulphate at room temperature. After 30 min stirring at 4 °C the mixture was centrifuged at 19,000 × $g$ for 20 min and the supernatant mixed with saturated ammonium sulphate to a final concentration of 60%. This mixture was stirred for 30 min at 4 °C then centrifuged for 20 min at 19,000 × $g$. The supernatant was mixed further with ammonium sulphate to a final concentration of 80%. After 30 min stirring at 4 °C the mixture was centrifuged for 30 min at 19,000 × $g$ then the pellet resuspended in 5 mL 10 mM MES buffer pH 6. The resuspended pellet was then loaded onto a 5 mL Fast Flow HiTrap SP column (GE Healthcare) equilibrated in 25 mM MES.NaOH pH 6 and the protein eluted with a linear gradient of 0–500 mM NaCl. Fractions containing YpbE$_{130-240}$ were pooled, concentrated and loaded onto a Superdex 75 XK16/60 gel filtration column (GE Healthcare) pre-equilibrated in PBS.

(v) YrrS; YrrSΔ$_{13-16}$ was expressed in BL21(DE3) cells by overnight induction at 20 °C with 0.4 mM IPTG. The pellet from a 2 L culture was resuspended in 25 mL 50 mM Tris.HCl (pH 8), 500 mM NaCl supplemented with 0.5 mg$^{-1}$mL$^{-1}$ lysozyme, 2 µg$^{-1}$mL$^{-1}$ DNAase and Roche Complete protease inhibitor cocktail. The cell suspension was lysed by sonication, and centrifuged at 10,000 × $g$ for 20 min at 4 °C to remove intact cells. The supernatant was centrifuged at 100,000 × $g$ at 4 °C for 1 h and the membrane pellet was resuspended in 15 mL 50 mM Tris.HCl (pH 8), 500 mM NaCl 10% (vol per vol) glycerol and centrifuged again for 1 hr at 100,000 × $g$ at 4 °C. The membrane pellet was frozen in liquid nitrogen and stored at −80 °C. The membrane pellet was resuspended in 10 mL of 50 mM Tris.HCl (pH 8), 500 mM NaCl, 2% reduced Triton X-100, 5 mM imidazole supplemented with Roche Complete EDTA free protease inhibitor cocktail. After 3 h gentle agitation at 4 °C the membrane suspension was centrifuged at 4 °C at 100,000 × $g$ for 30 min. A 0.4 mL bed volume of Ni-NTA Sepharose (Qiagen) was added to the supernatant and mixed overnight at 4 °C in a tube roller before filtering through an empty gravity column. The Ni-NTA beads were washed three times with 4 mL of 50 mM Tris.HCl (pH 8), 500 mM NaCl, 0.1% reduced Triton X-100, 5 mM imidazole and YrrSΔ$_{13-16}$ was eluted from the column with 4 mL of the same buffer supplemented with 250 mM imidazole. The eluate was concentrated to 0.5 mL and loaded onto a Superdex 200 HR10/300 gel filtration column (GE Healthcare) equilibrated in 10 mM Tris.HCl (pH 8), 250 mM NaCl, 0.1% reduced Triton X-100.

All recombinant proteins or peptides were concentrated and flash frozen in small aliquots in liquid nitrogen and stored at −80 °C.

**Crystallisation and structure determination.** Successful crystallisation of the N-terminal domains of GpsB proteins was improved by truncation of a few amino acids from both N- and C-termini, which are too far from the PBP binding site (25 and 45 Å, respectively) to affect binding and correspond to the minimum ordered portion of GpsB[19]. Co-crystallisation of $Bs$GpsB$_{5-64}$:$Bs$PBP1$_{1-17}$, $Bs$GpsB$_{5-64}$$^{Lys32Glu}$:$Lm$PBPA1$_{1-15}$ and $Sp$GpsB$_{4-63}$:$Sp$PBP2a$_{27-40}$ followed the same procedure. Equal volumes of GpsB protein and PBP peptide, both in a buffer of 10 mM HEPES.NaOH (pH 8.0) and 100 mM NaCl, were mixed at final concentrations of 20 and 25 mg$^{-1}$mL$^{-1}$, respectively. This mixture, corresponding to a 1:5 molar ratio of protein:peptide, was subjected to sparse matrix crystallisation screening at room temperature with 100 nL drops dispensed by a Mosquito (TTP Labtech) liquid handling robot into 96 well MRC crystallisation plates. Unbound $Sp$GpsB$_{4-63}$ was crystallised by the same procedure, at the same protein concentration, except for the absence of added peptide. Crystals of $Bs$GpsB$_{5-64}$:$Bs$PBP1$_{1-17}$ grew from 0.1 M HEPES/MOPS pH 7.5, 12.5% MPD, 12.5% PEG 1000 12.5% PEG 3350, 0.03 M MgCl$_2$, 0.03 M CaCl$_2$ and were mounted in rayon loops and frozen directly in liquid nitrogen. Crystals of $Bs$GpsB$_{5-64}$$^{Lys32Glu}$:$Lm$PBPA1$_{1-15}$ grew from 0.1 M imidazole

pH 8, 0.2 M zinc acetate, 20% PEG 3000 were transferred to a cryoprotectant comprising the well solution supplemented with 20% vol per vol PEG 300 and 2.5 mg$^{-1}$ mL$^{-1}$ of the $Lm$PBPA1$_{1-15}$ peptide; after 3 min the crystals were mounted in rayon loops and frozen in liquid nitrogen. The $Sp$GpsB$_{4-63}$:$Sp$PBP2a$_{27-40}$ complex crystallised in 0.1 M Tris.HCl pH 8.5, 0.2 M lithium sulphate, 40% PEG400; crystals were mounted after 1 μL of the crystallisation mother liquor containing 2.5 mg$^{-1}$ mL$^{-1}$ $Sp$PBP2a$_{27-40}$ peptide was layered over the crystallisation drop immediately prior to freezing directly in liquid nitrogen. Crystals of unbound $Sp$GpsB$_{4-63}$ were obtained from 0.1 M Tris.HCl pH 8.5, 0.2 M magnesium chloride, 20% PEG 8000 and were cryoprotected by direct transfer to a solution of the mother liquor supplemented with 20% PEG400 before mounting and freezing as above.

X-ray data were collected from crystals of unbound $Sp$GpsB$_{4-63}$ in house with a gallium METALJET$^{TM}$ (Bruker AXS GmbH) as X-ray source at a wavelength of 1.34 Å. All other diffraction data were collected at the DIAMOND synchrotron: for $Bs$GpsB$_{5-64}$:$Bs$PBP1$_{1-17}$ data were collected on beamline I24 at a wavelength of 0.9686 Å; for $Bs$GpsB$_{5-64}$$^{Lys32Glu}$:$Lm$PBPA1$_{1-15}$ data were collected on beamline I04 at a wavelength of 0.9795 Å and for $Sp$GpsB$_{4-63}$:$Sp$PBP2a$_{27-40}$ data were collected on beamline I03 at a wavelength of 0.9762 Å. For $Bs$GpsB$_{5-64}$:$Bs$PBP1$_{1-17}$ and $Sp$GpsB$_{4-63}$:$Sp$PBP2a$_{27-40}$ the diffraction images were indexed and integrated with XDS[71] and scaled with AIMLESS[72]. For $Bs$GpsB$_{5-64}$$^{Lys32Glu}$:$Lm$PBPA1$_{1-15}$ the diffraction images were indexed and integrated with DIALS[73], scaled in XDS[71] and merged with AIMLESS[72]. For unbound $Sp$GpsB$_{4-63}$ the images were indexed, integrated, scaled and merged in Proteum 3 (Bruker AXS GmbH). For the three different GpsB:PBP complexes and the unbound $Sp$GpsB$_{4-63}$ the initial phases were obtained by molecular replacement (MR) in PHASER[74]. For $Bs$GpsB$_{5-64}$:$Bs$PBP1$_{1-17}$ residues 5–64 from the structure of dimeric GpsB$_{1-68}$ (PDBid 4UG3[19]) were used as search model for MR whereas for $Bs$GpsB$_{5-64}$$^{Lys32Glu}$:$Lm$PBPA1$_{1-15}$ the MR search model was the co-ordinates of the $Bs$GpsB$_{5-64}$:$Bs$PBP1$_{1-17}$ complex with the peptide removed. For the unbound $Sp$GpsB$_{4-63}$ the search model for MR was the N-terminal domain of $Lm$GpsB (PDB code 4UG1[19]). For the $Sp$GpsB$_{4-63}$:$Sp$PBP2a$_{27-40}$ complex, the co-ordinates of the unbound $Sp$GpsB$_{4-63}$ protein were used as a search model for MR. In all cases the molecular replacement solutions were visualised and rebuilt in COOT[75]. The $Bs$GpsB$_{5-64}$$^{Lys32Glu}$:$Lm$PBPA1$_{1-15}$ and $Sp$GpsB$_{4-63}$: $Sp$PBP2a$_{27-40}$ complexes were refined in PHENIX.REFINE[76] while the $Bs$GpsB$_{5-64}$: $Bs$PBP1$_{1-17}$ complex and unbound $Sp$GpsB$_{4-63}$ were refined using REFMAC[77]. The Ramachandran plot statistics (favoured/allowed) are: $Bs$GpsB$_{5-64}$:$Bs$PBP1$_{1-17}$ 100%/ 0%; $Bs$GpsB$_{5-64}$$^{Lys32Glu}$:$Lm$PBPA1$_{1-15}$ 99.2%/0.8%; $Sp$GpsB$_{4-63}$:$Sp$PBP2a$_{27-40}$ 98.6%/ 1.4%, and unbound $Sp$GpsB$_{4-63}$ 99.6%/0.4% allowed. Statistics for data collection and for the final refined models can be found in Table 1. Representative electron density

maps for the peptide-bound structures are displayed in stereographic mode in Supplementary Figure 8.

**FP assays.** FP experiments were undertaken in a buffer of 10 mM Tris.HCl (pH 8.0), 250 mM NaCl, 0.1% reduced Triton X-100. The excitation wavelength was 540 nm and fluorescence emission was recorded above 590 nm for TAMRA-labelled peptides. The N-terminal domains of GpsB proteins $Bs$GpsB$_{1-68}$, $Lm$GpsB$_{1-73}$, $Sp$GpsB$_{1-63}$ were used in all experiments in preference to the full-length GpsB proteins due to the high solubility of the former, which facilitated achieving the high protein concentrations necessary to saturate peptide binding. The validity of the N-terminal domain as a substitute for full-length GpsB proteins in these experiments is supported by the comparable affinity of $Bs$GpsB and $Bs$GpsB$_{1-68}$ for a labelled $Bs$PBP1A$_{1-32}$ peptide in FP experiments (Supplementary Table 1). $K_d$ values ± the standard error, generated after simultaneously fitting all the binding data to a 1:1 interaction model are reported and there are three measurements at each data point in a titration.

**SPR.** All SPR experiments used a running buffer of 10 mM Tris.HCl (pH 8.0), 250 mM NaCl, 0.1% reduced Triton X-100. $Bs$PBP1 and $Bs$PBP1$_{17-914}$ were immobilised on the surface of a CM5 chip (GE Healthcare). For the $Bs$PBP1/YrrSΔ$_{13-16}$ titration 800 RUs of $Bs$PBP1 were immobilised on the chip surface; for the $Bs$PBP1/ $Bs$GpsB titration 1200 RUs of $Bs$PBP1 were immobilised. YrrSΔ$_{13-16}$, which lacks residues 13–16 in the cytoplasmic domain, was used in these experiments to minimise non-specific interactions with the chip matrix; binding of YrrSΔ$_{13-16}$ to the reference surface was negligible and the deletion of residues 13–16 leaves the GpsB-binding motif intact. The YrrSΔ$_{13-16}$ titration was carried out in single-cycle mode, without regenerating the surface between successive injections of protein, because harsh reagents were necessary for full surface regeneration. The fit of the single-cycle mode data to a 1:1 binding model was achieved with a $U$-value of 2 and a corresponding $\chi^2$ of 46.1.

**CD analysis.** CD spectra were recorded on a JASCO J-810 spectropolarimeter with a PTC-4235 Peltier temperature controller using 1 mm path length quartz cuvettes. For full wavelength scans a scan speed of 10 nm$^{-1}$ min$^{-1}$ and a response time of 4 s were used, the final spectra were the average of at least four measurements. $Bs$PBP1$_{1-32}$ CD spectra were recorded at 4 °C in a buffer of 20 mM sodium phosphate, pH 7.3 at a peptide concentration of 25 μM. Peptide helical contents were estimated from molar ellipticities at 222 nm using the equation $_{helix} = [\theta]_{222\ nm} / (39,500(1-2.57/n))$[78] where

## Table 1 Data collection and refinement statistics

| | $Bs$GpsB$_{5-64}$:$Bs$PBP1$_{1-17}$ | $Bs$GpsB$_{5-64}$$^{Lys32Glu}$:$Lm$PBPA1$_{1-15}$ | Unbound $Sp$GpsB$_{4-63}$ | $Sp$GpsB$_{4-63}$:PBP2a$_{27-40}$ |
|---|---|---|---|---|
| **Data collection** | | | | |
| Space group | $P\ 2_1\ 2_1\ 2_1$ | $P\ 1\ 2_1\ 1$ | $P\ 1\ 2_1\ 1$ | $C\ 1\ 2\ 1$ |
| $a, b, c$ (Å) | 31.5, 53.8, 85.9 | 26.6, 31.4, 81.0 | 39.1, 54.5, 61.6 | 83.1, 26.4, 65.9 |
| $\alpha, \beta, \gamma$ (°) | 90, 90, 90 | 90, 92.4, 90 | 90, 106.1, 90 | 90, 106.4, 90 |
| Resolution (Å) | 45.57–1.95 (2.00–1.95)$^a$ | 31.42–1.60 (1.63–1.60)$^a$ | 27.22–1.90 (1.94–1.90)$^a$ | 39.85–1.80 (1.84–1.80)$^a$ |
| $R_{pim}$ | 0.059 (0.473) | 0.044 (0.386) | 0.041 (0.311) | 0.051 (0.269) |
| CC (1/2) (%) | 99.4 (72.0) | 99.7 (74.3) | 99.8 (78.0) | 97.6 (88.3) |
| $I/\sigma I$ | 8.8 (2.1) | 8.4 (1.8) | 13.9 (4.0) | 9.0 (2.2) |
| Completeness (%) | 100.0 (100.0) | 99.8 (99.9) | 100 (100) | 99.8 (99.9) |
| No. of observations | 78,385 (5459) | 63,406 (3019) | 497,944 (28,050) | 46,904 (2454) |
| No. of unique reflections | 11,206 (768) | 17,944 (902) | 19,776 (1353) | 13,052 (763) |
| Redundancy | 7.0 (7.1) | 3.5 (3.3) | 25.2 (20.7) | 3.6 (3.2) |
| **Refinement** | | | | |
| Resolution (Å) | 45.57–1.95 | 29.29–1.60 | 26.01–1.90 | 39.85–1.80 |
| No. of reflections | 11,162 | 17,912 | 19,738 | 13,007 |
| $R_{work}/R_{free}$ | 0.187/0.214 | 0.173/0.204 | 0.188/0.249 | 0.178/0.225 |
| **No. of atoms** | | | | |
| Protein | 951 | 986 | 1919 | 999 |
| Peptide | 94 | 20 | — | 220 |
| Ligand/ion | 1 | 5/8 | — | 30/1 |
| Water | 48 | 106 | 260 | 87 |
| **B-factors (Å$^2$)** | | | | |
| Protein | 34.0 | 25.2 | 24.8 | 18.7 |
| Peptide | 39.3 | 36.0 | — | 32.5 |
| Ligand/ion | 23.7 | 30.9/36.5 | — | 57.9/12.1 |
| Water | 36.4 | 35.6 | 30.6 | 28.1 |
| **R.m.s. deviations** | | | | |
| Bonds (Å) | 0.006 | 0.016 | 0.017 | 0.015 |
| Angles (°) | 0.673 | 1.372 | 1.399 | 1.393 |

$^a$Values in parentheses refer to the highest resolution shell

$n$ is the number of residues in the peptide. $Lm$PBPA1$_{1-15}$ CD spectra were analysed at 40 μM in 20 mM sodium phosphate buffer, pH 7.8, for samples containing 0, 40 and 60% trifluoroethanol (TFE). For samples containing 80% TFE, 5 mM sodium phosphate buffer, pH 7.8, was used. The CD spectra of $Bs$GpsB$_{1-68}$ proteins were measured at a concentration of 4.5 μM in a buffer of 20 mM sodium phosphate (pH 7.8) 150 mM NaCl. For thermal melt experiments, a response time of 8 s and scan rate of 1 °C$^{-1}$min$^{-1}$ were used. All thermal melts were reversible as CD spectra recorded before and after the melts in the wavelength range 200–240 nm could be superimposed.

**Peptidoglycan pulldown assay.** PG pulldown assays were carried out in PBS buffer. *B. subtilis* PG (SigmaAldrich) was prepared as a 10 mg$^{-1}$ mL$^{-1}$ stock solution in PBS containing 0.02% sodium azide. 25 μg of protein were added to a 100 μL sample containing 66 μg of PG and the mixture incubated with gentle agitation at room temperature for 30 min. The mixture was centrifuged at 16,000 × $g$ for 7 min, the supernatant removed and the pellet resuspended in 1 mL of ice-cold PBS and immediately centrifuged for 2 min at 16,000 × $g$. The pellet was resuspended in 1 mL of the appropriate buffer and centrifuged for 2 min at 16,000 × $g$. Finally the pellet was boiled for 3 min in 60 μL SDS-PAGE loading buffer before analysis by SDS-PAGE; the SDS-PAGE loading buffer lacked reducing agents for YpbE$_{130-240}$$^{Ser132Cys}$ samples.

## Data availability

The crystallographic data that support the findings of this study are available in the PDB with the identifiers: 6GP7 for $Bs$GpsB$_{5-64}$:$Bs$PBP1$_{1-17}$; 6GPZ for $Bs$GpsB$_{5-64}$$^{Lys32Glu}$:$Lm$PBPA1$_{1-15}$; 6GQN for $Sp$GpsB$_{4-63}$:$Sp$PBP2a$_{27-40}$; and 6GQA for unbound $Sp$GpsB$_{4-63}$. All the other data that support the findings of this study are available from the corresponding author upon request.

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

## Acknowledgements

This work was supported by grants from the UK BBSRC (BB/M001180/1 to R.J.L.), the German DFG (HA 6830/1-1 and HA 6830/1-2 to S.H.), the American NIH (RO1GM113172, RO1GM114315 and RO1GM127715 to M.E.W.) O.M. is funded by a Contribution to Basic Research (LR7/2007) from the Autonomous Region of Sardinia. F. C. is funded by the European Commission (International Training Network Train2-Target, No. 721484) and Z.R. is funded by a UK BBSRC DTP studentship (BB/M011186/1). We thank the Diamond synchrotron light source for access to its beamlines and thank its staff for support during data collection. We are indebted to Waldemar Vollmer for critical reading of the manuscript and for supplying *S. pneumoniae* genomic DNA. We thank Helen Waller for technical assistance with SPR experiments, Dalia Denapaite, Regine Hakenbeck and Reinhold Bruckner for the supply of affinity-purified anti-*Sp*PBP2x antibodies, Jiaqi Zheng and Amilcar J. Perez for co-IP analysis with the anti-*Sp*PBP2x and anti-*Sp*GpsB antibodies, and Simon Thorpe at the University of Sheffield for mass spectroscopy analysis of peptide samples.

## Author contributions

R.M.C. and Z.J.R. conducted the biochemical and X-ray crystallography experiments and R.M.C., Z.J.R. and R.J.L. analysed these data. R.M.C. performed bioinformatics analyses. J.R. and S.H. conducted the Listeria BACTH screen and the fosfomycin resistance assays and analysed both these data. F.A.A. conducted the Bacillus BACTH screen, and F.A.A. and R.A.D. analysed these data. F.C. conducted the Streptococcus BACTH screen and F. C. and O.M. analysed these data. H.-C.T.T. conducted the Streptococcus phenotype and growth analyses and performed the Western experiments and H.-C.T.T., O.M. and M.W. E. analysed these data. R.M.C., H.-C.T.T, S.H., O.M. and R.J.L. wrote the first draft of the paper and all authors contributed to the final submitted manuscript.

## Additional information

**Competing interests:** The authors declare no competing interests.

