## [Peer Review File · Nature Communications]

Reviewers' comments:

Reviewer #1 (Remarks to the Author):

The article by Cleverley et al. tackles the structural and functional study of GpsB from three different bacteria, *B. subtilis*, *L. monocytogenes*, and *S. pneumoniae*, and describes efforts to understand its interaction partners. The work seems to be an extension of work previously obtained by some of the authors and published in *Mol Mic* in 2015; that manuscript included the first structure of GpsB from *B. subtilis* and notable data on the interaction with PBP1a. Despite this manuscript being of interest, there are several weak points and many assumptions that are problematic.

Lines 109-121: in this section of the text, authors painstakingly list the interactions between the two peptides that can actually be directly seen in Figure 1. It is unclear to this reviewer why the interactions should be listed and described, considering that they are visible in the figure. In addition, if the complex can only be seen between one GpsB and one PBP1 peptide, why is a GpsB dimer shown?

Line 99: The claim that interactions 'drive' association generally presupposes that there is an element of catalysis or energetic exchange involved in the interaction. This is not the case here; these are simply interactions between two peptides. The term 'drive' is thus not appropriate for the description.

Lines 128-129: '... BsPBP11-32Arg28Ala had no effect on binding, confirming that non-specific electrostatics do not drive the BsGpsB:BsPBP1 interaction'. This sentence makes no sense. What are 'non-specific electrostatics'? By modifying the amino acid from an arginine to an alanine, authors did much more than block the possibility of electrostatic interactions; the entire carbon-based section of the Arg chain was also removed.

Line 133-134: it is unclear why this statement is important. Different residues can serve as helix cappers in peptides, and the statement seems to fit well in a biophysical paper but not in one where the reader is trying to understand the biological function(s) of GpsB.

Lines 140-142: this statement suggests that the absence of GpsB only seems to have an effect on the cell if 'unnatural', laboratory conditions are employed. Can authors comment on this?

Lines 148 – 168: authors then attempted to obtain crystals of the analogous complex from *Listeria monocytogenes*, but were unable to do so. Thus, they crystallized and solved the structure of a hybrid complex, using GpsB from *B. subtilis* and PBP1a from *L. monocytogenes*. Most of the complex was disordered, except for a core region, shown in Fig. 2A. Nevertheless, authors went on and did affinity experiments between the proteins from the two different species. From this section, which is the weakest in the paper, it seems like the only take home message is that Arg8 from the PBP1 peptides from the two species play an important role in the interaction.

line 218: a K_d of 430 μ M is very high, leaving the reader wondering what the relevance of this interaction really is.

Lines 226-227 and Figure 3: the BACTH results do not show positive or negative controls.

Figure 3d: despite the fact that authors attempted to make a model that would suggest how the different interactions could come about, this figure is not helpful. What do the different shapes refer to? The figure does not seem to be to scale, but if the objective is to make a model, it should at least attempt to be (or residue numbers should be indicated).

Lines 235-252: authors here attempted to understand if the same findings could be applied to the

Listeria monocytogenes system, but homologs of YpbE/YrrS could either not be found or those identified did not present the desired sequences. They then set out to test other well-known division/elongation experiments by BACTH. Suppl figure 3, where they present the latter results, does not display any controls, positive or negative; careful controls, including proteins not involved in division/elongation and that are expected/not expected to interact, should be done in the same conditions used by the authors (incubation at 30C for 1, 2 days)

Lines 302-304: it is not surprising that the deletion of somewhat long stretches of amino acids (one of which is 21 aa long) in a key protein would generate morphological defects.

Line 333-334: authors mention that '...The PG synthase arginine 'finger' pokes into a negatively-charged cavity situated between α -helices 1 and 2 of GpsB and is fixed in the same orientation in all structures ...' this phrase suggests a general nature for the findings, which is not completely factual, since the structural information for the Listeria system was obtained from the structure of a hybrid complex.

Lines 356-358: here authors highlight an important point: their GpsB seems to interact with a large number of proteins, at least in their BACTH. How/why a cytoplasmic peptide would do this remains unknown, but this brings to light the fact that all of the interaction data were obtained by BACTH, a technique that is known to generate false positives/false negatives. Given the fact that this was the only interaction technique employed in the manuscript, all of these interactions should either be taken with a grain of salt, or confirmed by other techniques.

Lines 394-427: sections of this text read like a review article, and describe supposed interactions.

Minor comments:

The text refers to different forms of the proteins without any explanation. For example, GpsB from *B. subtilis* is referred to as GpsB, GpsB5-64, GpsB1-68, and GpsB1-63 (the latter in the figure). In the crystallization table, only BSGpsB5-64 is mentioned. It is unclear if there were some typos, or if these are different constructs. PBP1 is sometimes also referred to as PBP1 1-32. Is this the same thing?

Fig.1b does not allow the reader to fully grasp how this complex is being formed. A figure where GpsB is shown as a surface and the N-terminal region of the PBP is shown as a helix would allow a much better visualization.

Lines 144-146: '... The cytoplasmic mini-domain of LmPBPA1 has an abundance of positively charged residues (Supplementary Figure 2A) ...': This is not evident from the figure, residues should be highlighted

'...but lacks an exact copy of the SRxxR(R/K) motif of Bacillaceae PBP1 (Supplementary Figure 1A)...': It is unclear how FigS1a highlights this issue

Line 177: the expression 'in vivo' is more often employed for work with animals; here, authors have performed microbiological experiments, or 'in cellulo'.

Reviewer #2 (Remarks to the Author):

In their manuscript „The cell cycle regulator GpsB functions as cytosolic adaptor for multiple cell wall enzymes“ Cleverley et al. complement previous findings on the cellular roles of GpsB,

providing a structural basis for the interaction of the cytoplasmic cell cycle regulator with PBP's and thus provide mechanistic insights in the organization and coordination of the cell wall synthesis machinery. The authors identified the crucial sequence motif, involving a conserved Arg residue in class A PBP's and relevant residues in GpsB for different bacteria. A comprehensive structural analysis of PBP:GpsB pairs using PBP cytoplasmic mini-domains enlightened crucial interaction sites supported by a mutagenesis approach allowing for the detailed binding motif analysis, determination of binding constants and phenotypic analysis. Based on this knowledge additional proteins are predicted to be members of the GpsB interactome. The authors provide a comprehensive set of data and translate structural findings to the cellular level to support their conclusions. The research topic covered in this work is of major importance for the field and highlights the complexities of bacterial cell physiology and the complex enzymatic cell envelope networks. That said, although well written the manuscript is sometimes hard to "digest".

I suggest considering the following points:

Abstract, last sentence: I recommend to delete the statement regarding the design of antibiotics. There is no basis that supports such a statement, nor has this been discussed in the main text. Essentiality (not true for all GpsBs discussed) or being part of a larger protein complex involving essential proteins do not define a "good" antibiotic target. Moreover, in the past such target based approaches largely failed and the development of resistance towards such targets is more than likely.

Results, line 107: Can the authors comment on why BsGpsB5-64 has been used for structural analysis and BsGpsB1-68 (wt and variants) for FP and CD? The same is true for SpGpsB4-63 and SpGpsB1-63, respectively or LmPBPA11-20 and LmPBPA11-21.

Line 156: Is BsGpsB1-68 correct in this context (structure elucidation)? Shouldn't it read 5-64?

Line 150: An alignment of GpsB of the different bacteria used in this study may be helpful. Here it would be especially helpful to highlight GpsB interfacial residues.

Figure legend 2: Is LmPBPA11-15 correct? The text states (starting line 150): LmPBPA11-21 and LmPBPA11-15. This is very confusing.

Line 166: for all relevant amino acids of LmPBPA1 the KD values of respective mutant proteins have been determined, except for Thr7.

Line 246: exchange ",," to "."

Line 255: delete "and GpsB"

Line 279: self-interaction of GpsB may not indicate functionality.

Paragraph starting line 285: Given the enormous interaction studies for Bacillus and Listeria it is noticed that BacTH-analysis is lacking for *S. pneumoniae*.

Reviewer #3 (Remarks to the Author):

Cleverley, et al. report the crystal structures of a cell cycle regulator GpsB in complex with the cytoplasmic "mini-domains" of PBPs from three bacteria (*B. subtilis*, *L. monocytogenes*, and *S. pneumoniae*). The structures reveal motifs that are important for the protein-protein interactions, leading to identification of two new GpsB-interacting proteins in *B. subtilis*. The authors generated extensive amounts of data to analyze the protein-protein interfaces using FP and the bacterial two-hybrid system. They further provide some evidence to show significance of the protein-protein interfaces on the protein functions for cell growth and drug susceptibility. Overall, this is a solid structural biology paper demonstrating detailed interactions of cell wall synthases and the adaptor proteins in the low-GC Gram-positive bacteria.

Specific comments:

1. Are the BsGpsB residues important for binding to the PBP1 mini-domain critical for the cellular function of GpsB? This can be examined in *B. subtilis* *gpsB ezrA* or *gpsB ftsA* double mutant which shows poor growth with severe cell division.

2. The affinity of BsPBP1 to GpsB was 0.7 μ M in SPR (line 104) and no binding was observed at 25

uM BsPBP1 lacking the first 17 amino acids, indicating requirement of the 17 aa for the interaction. However, in the FP assay, the affinity of the BsPBP1(1-32) peptide was low at 120 uM (Supplementary Table 1). I think this is disconnect and could be due to affinity reduction by the fluorophore attached to PBP1 peptide or due to interactions between other domain of PBP1 and GpsB. An SPR experiment with the non-labeled PBP1 peptide should clarify this. If the affinity of the peptide is low in the SPR, the submicromolar affinity of GpsB and PBP1 could be an artifact because other PBP1 domain is in TM and the periplasm. In addition, since GpsB is a hexameric protein, what was the stoichiometry binding in the SPR data?

3. In Figure 2C, PBPA1 and GpsB interact each other and may work together, but why is the PBPA1 deletion mutant more resistant to fosfomycin than WT in *L. monocytogenes* while *gpsB* deletion was more sensitive to the antibiotic? In addition, is the PBP1A PBPA2 double deletion mutant more sensitive to fosfomycin? If so, is the drug sensitivity change of the double mutant comparable to the *pbpA1(Q10P)* PBPA2?

4. Line 39 in the abstract, the authors state that this could represent a starting point for the design of much needed new antibiotics. However, the manuscript does not include any description or discussion about how this could lead to a starting point to design new antibiotics. So this statement is unclear.

Minor comment:

The authors used a variety of peptides, proteins, and variants. It would be helpful to show clearly in figures what protein/peptide variants were used.

Dear Reviewer

Thank you for your time spent assessing our manuscript. We have striven to carefully revise it where possible, with further explanation provided point-by-point in the red text below. To summarise, the Results section has been updated quite substantially, new data have been incorporated and some of the figures have been remade and others have been added. In response to all 3 reviewers, we have (i) added additional fluorescence polarisation (FP) interaction data to SI Table 1 and present new data in SI Fig 6 on the interaction of *SpGpsB* with different cell division proteins studied by various methods, including FP, to address a perceived concern about the dependency of the study on BACTH data and to present a more expansive analysis of the interaction partners of GpsB in *S. pneumoniae*; (ii) defined each construct as they appear in the main text. In response to reviewers 2 and 3 we have removed the last sentence from the abstract. In response to reviewers 1 and 3 we have (i) explained the likely relevance of the interactions with apparently weak K_d values and how they will be affected by the avidity effect (lines 255-258, and Legend to SI Table 1). Furthermore, in response to reviewer 1, we have (i) added a new panel to Fig 1B to show the surface of GpsB in the *BsGpsB*₅₋₆₄:*BsPBP1*₁₋₁₇ complex; (ii) removed the word 'drive' (and 'driven') from the manuscript; (iii) clarified on lines 138-141 and 194-197 what we had tried to convey by 'non-specific electrostatics'; (iv) provided a better rationale on lines 162-171 for presenting the 'hybrid' complex; (v) explained how some of the constructs in Fig 3 act as internal positive and negative controls, with text clarifying this point introduced in the Legend to Fig 3C; (vi) adjusted Fig 3D to take into account the reviewer's points and updated its Legend; and (vii) explained better on lines 278-283 and in the Legend to SI Fig 3C how the data in SI Fig 3C represents only a part of a much larger screen against divisome/elongasome proteins. For reviewer 2, we have (i) added new SI Fig 1A to show an alignment of GpsB from the different bacteria used in this study; (ii) corrected minor typos; (iii) addressed the self-interaction / functionality concern on Lines 318-323 and 333-340; and (iv) extended the BACTH analysis in *S. pneumoniae* and included a new Figure (SI Fig 6) and new text in the Results and Discussion that also contains and compares interaction data obtained by other methods. For reviewer 3, we have (i) included some additional data within the rebuttal regarding the effect of the fluorophore (please see page 11) and refer to these data in the legend to Fig 1D and also provided text (lines 255-258, and Legend to SI Table 1) within the manuscript regarding the likely avidity effect of GpsB's interaction with membrane-embedded proteins; (ii) explained the fosfomycin sensitivity in the legend to Fig 2C.

All line numbers referred to in the author response sections correspond to the line numbers of the revised manuscript, and therefore differ slightly to the original line numbers cited. We have also uploaded a 'tracked-changes' version of the manuscript should it make life easier for you to follow how the manuscript has changed from the previous version.

Thank you

Rick Lewis, on the behalf of Cleverley et al

Reviewer #1 (Remarks to the Author):

The article by Cleverley et al. tackles the structural and functional study of GpsB from three different bacteria, *B. subtilis*, *L. monocytogenes*, and *S. pneumoniae*, and describes efforts to understand its interaction partners. The work seems to be an extension of work previously obtained by some of the authors and published in *Mol Mic* in 2015; that manuscript included the first structure of GpsB from *B. subtilis* and notable data on the interaction with PBP1a. Despite this manuscript being of interest, there are several weak points and many assumptions that are problematic.

Lines 109-121: in this section of the text, authors painstakingly list the interactions between the two peptides that can actually be directly seen in Figure 1. It is unclear to this reviewer why the interactions should be listed and described, considering that they are visible in the figure. In addition, if the complex can only be seen between one GpsB and one PBP1 peptide, why is a GpsB dimer shown?

Author response: First, we chose to describe the interaction in detail for one of the structures of the three GpsB-peptide complexes and to limit the description of the remaining two structures to the bare minimum to avoid repetition. We feel strongly that one complete description should be given to help the reader (especially those who are not structural biologists) understand the really important interactions between GpsB and its PBP binding partners, which involve a critical arginine in each case. Second, the N-terminal domain of GpsB is an obligate dimer and displaying it as a monomer would thus not make sense. We had provided a rationale (in the legend to Fig 1B) for the binding of only a single peptide in this particular instance, as one site is blocked by crystal contacts, but to make this point clearer we have simply moved this text from the legend to Fig 1B to form a new sentence on lines 126-128.

Line 99: The claim that interactions ‘drive’ association generally presupposes that there is an element of catalysis or energetic exchange involved in the interaction. This is not the case here; these are simply interactions between two peptides. The term ‘drive’ is thus not appropriate for the description.

Author response: Apologies for the confusion, and we have changed ‘drive’ to ‘govern’ here, and made similar changes elsewhere to a term that could not be mistaken for an active process.

Lines 128-129: ‘ BsPBP11-32Arg28Ala had no effect on binding, confirming that non-specific electrostatics do not drive the BsGpsB:BsPBP1 interaction’. This sentence makes no sense. What are ‘non-specific electrostatics’? By modifying the amino acid from an arginine to an alanine, authors did much more than block the possibility of electrostatic interactions; the entire carbon-based section of the Arg chain was also removed.

Author response: What we tried to convey by “non-specific electrostatics” was that the interaction between the PBP peptide and GpsB does not depend solely on the overall positive charge of the peptide irrespective of the position within it of positively-charged amino acids. Yes, the alkyl chain of the arginine is removed when replaced by alanine, but the *BsGpsB*₁₋₆₈:*PBP1*₁₋₃₂ interaction is unaffected by the Arg28Ala substitution (K_d of 90 μ M, vs 120 μ M for the same interaction using wildtype proteins and peptides, Supplementary Table 1). By contrast, the much more important residue for *BsGpsB*₁₋₆₈:*PBP1*₁₋₃₂ interactions, Arg8, sees a ~15-fold reduction in affinity when replaced by either alanine or lysine, which means that arginine specifically at position 8 is essential and not just positive charge and/or a long alkyl chain. We have thus modified the text on lines 136-141 and 194-197 in response to this point.

Line 133-134: it is unclear why this statement is important. Different residues can serve as helix cappers in peptides, and the statement seems to fit well in a biophysical paper but not in one where the reader is trying to understand the biological function(s) of GpsB.

Author response: This statement is important for rationalizing how GpsB interacts with PBP cytoplasmic mini-domains and is relevant to predicting other potential GpsB interaction partners. To clarify the importance of this statement we have adapted the text at lines 143-146. Regarding the biophysics, very few residues can actually act as helix N-cappers because the sidechain of residue *i* that N-caps the helix must accept a hydrogen bond from the amide nitrogen of residue *i*+3 in the

helix. There are numerous studies (from Baldwin, Doig, Thornton, Rose and the Richardsons, amongst others) over the last ~30 years that all agree that the most prevalent N-caps are Ser, Thr and Asn because each can accept the necessary hydrogen bond from the amide nitrogen and these sidechains can adopt the correct rotamer and, crucially, are the right length to form this essential H-bond. Longer sidechains (e.g. Gln, Glu, His) that could, in theory, still accept the H-bond are rare N-caps because they are too long to accept the H-bond without adopting unusual and thermodynamically disfavoured sidechain rotamers. Since sequence alignments of PBP1 proteins show that the first position in the motif is almost always Ser (Supplementary Fig. 1E) it is necessary to explain the sequence conservation in relation to the binding of PBP1 to GpsB. In this case, the N-capping of the helix by Ser7 is an important binding consideration (replacing the N-cap Ser with alanine reduces affinity ~6-fold, Supplementary Table 1) because of its helix stabilisation (Supplementary Figure 1D) rather than by forming a direct contact with GpsB. Therefore, we prefer not to change the manuscript further in this context.

Lines 140-142: this statement suggests that the absence of GpsB only seems to have an effect on the cell if ‘unnatural’, laboratory conditions are employed. Can authors comment on this?

Author response: We actually did comment on the conditional lethality of *gpsB* in *B. subtilis* and *L. monocytogenes* and the essentiality of *gpsB* in *S. pneumoniae* (original lines 419-422, new lines 504-507 in the Discussion), and suggested that redundancy in the larger genomes of bacilli and listeria in comparison to the pneumococci may explain this phenomenon.

Lines 148 – 168: authors then attempted to obtain crystals of the analogous complex from *Listeria monocytogenes*, but were unable to do so. Thus, they crystallized and solved the structure of a hybrid complex, using GpsB from *B. subtilis* and PBP1a from *L. monocytogenes*. Most of the complex was disordered, except for a core region, shown in Fig. 2A. Nevertheless, authors went on and did affinity experiments between the proteins from the two different species. From this section, which is the weakest in the paper, it seems like the only take home message is that Arg8 from the PBP1 peptides from the two species play an important role in the interaction.

Author response: An important “take home” message from the study of *LmGpsB:LmPBPA1* interactions is that, as with the *BsGpsB:BsPBP1* complex, the alpha-helical propensity of the *LmPBPA1* cytoplasmic mini-domain is an important determinant of GpsB binding and in both cases it is implicit from the structural and binding analyses that the critical arginine is at the start of the relevant helix. We hope that the reviewer feels that the adjustments we have made to the text in this section of the results now emphasize this more clearly. Regarding the relevance of *BsGpsB*^{K32E} as a surrogate for *LmGpsB*, every residue bar one (K32E) within 8 Å of the PBP1 binding site in *BsGpsB* is maintained in *LmGpsB*. Therefore, it is not just those residues that interact directly with the peptide that are maintained but also ‘bystander’ residues that help indirectly to shape the PBP binding site. The measured affinity of the *LmPBPA1* peptide for the K32E variant of *BsGpsB*₁₋₆₈ is 190 μM, whereas for the same peptide has an affinity of 200 μM for its cognate *LmGpsB*₁₋₇₃ (Supplementary Table 1). We have made changes to the text on lines 160-171 to clarify the important justifications for working with the “hybrid” complex. We chose to perform the fluorescence polarisation (FP) experiments on the “hybrid” complex so as to be consistent with the crystallographic studies. The distribution of blue intensities in the *Lm* BACTH, Fig 2B, is consistent with the affinity measurements in Supplementary Table 1: R8 is critical for the interaction, R12 slightly less so and Y11 slightly less so again.

It is also clear from the phenotyping data in Figure 2C, Supplementary Figure 2D that the Arg8Ala mutation in *LmPBPA1* had negligible effect *in vivo* yet it reduced binding by >15-fold *in vitro*. Thus, whilst Arg8 is important it is not the only consideration. We prefer to include the data on *LmGpsB* as it helps define common and species-specific functions of GpsB, one of our stated

ambitions for this project, but have made substantial changes to the text in this section of the paper that we hope explains its relevance more clearly.

line 218: a K_d of 430 μM is very high, leaving the reader wondering what the relevance of this interaction really is.

Author response: Please also see our response to reviewer 3, point 2. Yes, 430 μM is a high K_d value, but we are confident in our data, first because the error bars on the experiment are small, <5% of the absolute value, and in line with the errors of all the other FP measurements reported. Second, the interaction is specific since mutation of two GpsB residues that play important roles in binding to PBP peptides in all 3 structures described and for the interaction with YpbE (Fig 3A) abrogate binding to YrrS completely (compare the sigmoidal curve of the WT interactions to the straight lines of the Y25F and D31A mutants in Fig 3B). If the GpsB:YrrS interaction was simply the result of non-specific binding, then it seems hardly likely that the loss of a single hydroxyl in the Y25F mutant would have such a drastic effect. Third, the FP measurements represent the binding of one (or two) peptides to the dimeric N-terminal domains of GpsB in solution and therefore do not take into account the potential for avidity effects when the full-length GpsB hexamer interacts with membrane-localised partner proteins *in vivo*. Avidity effects would result from simultaneous interaction of GpsB with YrrS and the cell-membrane and/or simultaneous interaction of one GpsB hexamer with two membrane-localised proteins (for example with two YrrS molecules or with one YrrS molecule and a PBPI molecule).

Perhaps in hindsight, and our sincere apologies to the reviewers, we did not explain terribly well in the submitted manuscript that the measurements made by FP *in vitro* do not necessarily recapitulate well the situation *in vivo*: YpbE, YrrS and the PBPs are all integral membrane proteins and their movement in the membrane is thus restricted to a 2-D plane. As GpsB associates with and YrrS integrates into membranes, the YrrS cytoplasmic mini-domain and its cognate binding site on GpsB will be brought into close proximity, effectively increasing the local concentration of each significantly; this is the avidity effect. In this regard it is pertinent to note that the interaction of colicins with their cellular *E. coli* receptors have been measured *in vitro* (in solution with detergent-solubilized proteins) by many groups and are usually found to be micromolar or higher, yet it is also well known that a single colicin molecule is sufficient to kill an *E. coli* cell, implying that the interaction *in vivo* is much tighter. In Johnson *et al* (doi:10.1111/mmi.12568), ColN bound to the surface lipid LPS with an affinity of 2 μM , which is enough to bind to the bacterium and kill it, and binding was dependent upon the sugar moiety of LPS. However, when the lipid acyl chain was removed from LPS and binding of ColN to the free oligosaccharide in solution was assessed, no binding could be detected, indicating an avidity effect to due to simultaneous binding of the colicin with both the sugar and the acyl moiety of the LPS. Therefore, the measurements of affinities in solution of components that would ordinarily only interact in the context of a membrane can be misleading. We have thus introduced new text on lines 255-258 and in the Legend to Supplementary Table 1, with the inclusion of a citation to Johnson *et al.*, to rationalise the K_d values reported. For clarity regarding the FP data we have also added new data to Supplementary Table 1 and new text to the Methods (lines 599-605) that describe how the FP measurements were made to illustrate that the N-terminal domain of GpsB is a valid substitute for full length GpsB in the FP experiments.

Lines 226-227 and Figure 3: the BACTH results do not show positive or negative controls.

Author response: Our apologies, we did not make clear how the T25-GpsB₆₆₋₉₈ construct serves as an internal control for the following reasons. The lack of any interaction of the GpsB₆₆₋₉₈ fusion with all non-GpsB partner proteins is a valid negative control, particularly in the case of YrrS and YpbE, since the PBP-like GpsB binding motifs in both proteins bind solely to the N-terminal

domain of GpsB (Fig 5A, 5B). The C-terminal domain of GpsB also does not interact with any other partner proteins in the *L. monocytogenes* BACTH screen (please see Supplementary Fig 3C). The T25-GpsB₆₆₋₉₈ construct interacts with full length GpsB in the BACTH – itself a valid interaction given that GpsB is a hexamer (Rismondo et al 2016, Cleverley et al 2016). We have added extra sentences to the legend to Figure 3C in clarification of the role of T25-GpsB₆₆₋₉₈ as an internal control.

Figure 3d: despite the fact that authors attempted to make a model that would suggest how the different interactions could come about, this figure is not helpful. What do the different shapes refer to? The figure does not seem to be to scale, but if the objective is to make a model, it should at least attempt to be (or residue numbers should be indicated).

Author response: We have updated the legend to make clear which PDB file has been used for which protein, and have corrected the scaling error mistakenly introduced during the generation of the figure. Labelling the figure with residue numbers would be misleading since it would imply that the orientation of each protein relative to the others is known, which is not the case (except for GpsB:PBPA1 interactions). In all cases the membrane proteins in the figure adopt the same topology with respect to the membrane – the N-terminal domains are cytoplasmic and the C-terminal domains are extracytoplasmic and the legend has been changed to reflect this information. The residue numbers for the transmembrane regions has been added to the legend (despite several attempts they did not reproduce well in the figure itself). We have also now used a single colour for GpsB for simplicity. Therefore, both figure and legend have been updated, and we apologise for any unintended confusion.

Lines 235-252: authors here attempted to understand if the same findings could be applied to the *Listeria monocytogenes* system, but homologs of YpbE/YrrS could either not be found or those identified did not present the desired sequences. They then set out to test other well-known division/elongation experiments by BACTH. Suppl figure 3, where they present the latter results, does not display any controls, positive or negative; careful controls, including proteins not involved in division/elongation and that are expected/not expected to interact, should be done in the same conditions used by the authors (incubation at 30C for 1, 2 days)

Author response: The data shown in Supplementary Fig. 3C represent an extract from a much bigger screen that included 27 listerial components of the divisome and elangosome, and each protein in the screen was listed in the Methods in the Supplementary Information in the original submission. The self-interaction of T25-GpsB with GpsB-T18 serves as an excellent control since GpsB is a hexamer (Rismondo et al 2016, Cleverley et al 2016). T25 without a fusion partner was used, and was displayed in the figure, as a negative control (though it was not described as such, an oversight for which we apologise). Only the proteins that interacted with GpsB in the original screen were shown, and this point has now been made on lines 278-281. We have also added additional text to the Legend to Supplementary Fig. 3C.

Furthermore, the impact of point mutations in *Lm*GpsB (Y23A, D33A, D37A) on the interaction with PBPA1 correlates perfectly with our structural and FP binding data and therefore also serves as controls and supports further the relevance of the interactions detected by BACTH. The differential impact of these mutations on the interactions with certain partners – for example a D37A mutation affects the interaction with PBPA1 but not with MreC – also provides additional support that the interactions described are not artefacts of the technique. Please also note that in response to reviewer 2, line 285 we now report new BACTH data on the interaction of *S. pneumoniae* divisome/elongosome proteins with GpsB (and the equivalent alleles to the *Lm*GpsB variants in Supplementary Fig. 3C). The fact that the *Lm* and *Sp* BACTH data converge also supports our

conclusions on the make-up of the GpsB interactome.

Lines 302-304: it is not surprising that the deletion of somewhat long stretches of amino acids (one of which is 21 aa long) in a key protein would generate morphological defects.

Author response: The deletions have no effect on protein expression levels (Supplementary Fig 5C) and in all cases delete amino acids distant from the predicted start of the TM helix (residue 54). The deletions are thus unlikely to affect the insertion of *SpPBP2a* into the membrane but instead are likely the direct result of decreased interactions between *SpPBP2a* and GpsB. We have changed the text on lines 348-351 to clarify the location of the truncations in the domain. We agree that it is not surprising that deletions of up to 21 amino acids in the *SpPBP2a* cytoplasmic mini-domain generates morphological defects, as the truncations encompass the GpsB-binding motif in *SpPBP2a* and the impact of the various truncations in the BACTH data correlates well with the phenotypes of the deletion mutants *in vivo* (Supplementary Figure 5). What was quite surprising was that the deletion mutants that showed progressive decreased *SpPBP2a*:GpsB interactions were still viable, even in the absence of *SpPBP1a*, and did not phenocopy the Δ *gpsB* mutant. This is also true for *Listeria monocytogenes* and suggests that the Δ *gpsB* phenotype could be due to the loss of the interaction with another partner(s) in addition to *SpPBP2a*. Alternatively, mutating or deleting the GpsB binding motif in *SpPBP2a* doesn't abolish the interaction with GpsB completely or even reveals or generates an unexpected GpsB binding site perhaps in the juxta-membrane region that is still present in the deletion mutants. We have added text on lines 356-364 to address this point.

Line 333-334: authors mention that ‘...The PG synthase arginine ‘finger’ pokes into a negatively-charged cavity situated between α -helices 1 and 2 of GpsB and is fixed in the same orientation in all structures ...’ this phrase suggests a general nature for the findings, which is not completely factual, since the structural information for the *Listeria* system was obtained from the structure of a hybrid complex.

Author response: Please see above in relation to our response to this reviewer's comments about original lines 109-121 and 148-168.

Lines 356-358: here authors highlight an important point: their GpsB seems to interact with a large number of proteins, at least in their BACTH. How/why a cytoplasmic peptide would do this remains unknown, but this brings to light the fact that all of the interaction data were obtained by BACTH, a technique that is known to generate false positives/false negatives. Given the fact that this was the only interaction technique employed in the manuscript, all of these interactions should either be taken with a grain of salt, or confirmed by other techniques.

Author response: We agree that protein:protein interaction studies detected using a single technique, including BACTH, should be viewed cautiously and the biological significance of the observed interactions should be assessed rigorously. Unfortunately, the biochemical properties of many of the proteins identified in the screen are uncertain or unknown and hence the impact of their interaction with GpsB cannot be determined. We disagree with the assertion that all the interaction data in this study were obtained only by BACTH and that this technique is prone to generating false positives and negatives. First, BACTH does not, as commonly perceived, generate false results all that frequently, in fact, Battesti et al (doi: 10.1016/j.ymeth.2012.07.018) report that the occurrence of ‘sticky’ or ‘auto-activating’ proteins (*i.e.* proteins that bind to all others or the T18- and T25-fragments of adenylate cyclase) is actually a rare event. None of the T18/T25-hybrid proteins that we have tested interacted with the T18- and T25-domains alone, providing an additional control in the experiment. Second, the T18/T25-hybrid proteins often interacted differently with GpsB wildtype and allelic variants, also providing an additional control against false-positives. The allelic variants are particularly robust as controls since they retain the ability to self-interact or to interact

with certain other cell division proteins (for example in the *S. pneumoniae* BATCH screen an I36A mutation impacts interactions with PBP2a but not with PBP1a and doesn't impact self-interactions). Finally, we have also developed a FP assay (Figs 1D, 2A, 2B, 3A, Supplementary Figs 1B, 2B) and quantified the interactions between several N-GpsB domains and the cytoplasmic regions of 3 class A PBPs, YrrS and YpbE, and numerous mutated versions of N-GpsB and the 3 PBPs. Indeed, Table S1 lists 36 affinity measurements, and we have observed a strong correlation between the loss of interaction in BACTH and in FP. We have used SPR to study the dependency of the cytoplasmic min-domain of *Bs*PBP1 for its interaction with the N-terminal domain of *Bs*GpsB (Fig 1A), and also used SPR in kinetic mode to investigate the interaction of full-length YrrS with full-length PBP1 (Supplementary Fig 3B). Please also note that the greatly expanded data presented in SI Fig 6 is of particular interest because the interaction of *Sp*GpsB and allelic variants with PBP1a, PBP2b, PBP2x, EzrA and StkP obtained by BACTH can be rationalised against previously published (Rued et al 2017) and new co-IP data. For example the combined interaction data in Supplementary Figures 6A-6C together agree that the *Sp*GpsB:PBP2b interaction depends upon a bridging protein, identity currently unknown.

Lines 394-427: sections of this text read like a review article, and describe supposed interactions.

Author response: In the Discussion we tried to relate our new data to the state of knowledge in the cell division field in bacilli, listeria and pneumococci. We also drew parallels between GpsB and its function as an adaptor to other systems with similar adaptation functions (e.g. 14-3-3 proteins in eukaryotes) and where avidity effects are also likely to play a significant functional role (antibody/antigen interactions). Without clearer direction from the reviewer, it is difficult to see what else we should be doing here. Consequently, we have made few changes to this section.

Minor comments:

The text refers to different forms of the proteins without any explanation. For example, GpsB from *B. subtilis* is referred to as GpsB, GpsB5-64, GpsB1-68, and GpsB1-63 (the latter in the figure). In the crystallization table, only BSGpsB5-64 is mentioned. It is unclear if there were some typos, or if these are different constructs. PBP1 is sometimes also referred to as PBP1 1-32. Is this the same thing?

Author response: We had defined every construct used in the Supplementary Materials and Methods, however, to help reduce confusion we have now defined each one as it appears in the main text.

Fig. 1b does not allow the reader to fully grasp how this complex is being formed. A figure where GpsB is shown as a surface and the N-terminal region of the PBP is shown as a helix would allow a much better visualization.

Author response: We have added an additional panel to Fig 1B.

Lines 144-146: '... The cytoplasmic mini-domain of LmPBPA1 has an abundance of positively charged residues (Supplementary Figure 2A) ...': This is not evident from the figure, residues should be highlighted

Author response: We have now coloured the amino acids in the alignment by charge (blue for positive, red for negative).

'...but lacks an exact copy of the SRxxR(R/K) motif of Bacillaceae PBP1 (Supplementary Figure 1A)...': It is unclear how FigS1a highlights this issue

Author response: Apologies, this was a typo, we meant Supplementary Figure 1E)

Line 177: the expression 'in vivo' is more often employed for work with animals; here, authors have performed microbiological experiments, or 'in cellulo'.

Author response: With respect, *in cellulo* is not a term any of these authors recall encountering before, as a compromise we have changed 'in vivo' to 'in bacterial cells' and hope that this is acceptable to the reviewer.

Reviewer #2 (Remarks to the Author):

In their manuscript „The cell cycle regulator GpsB functions as cytosolic adaptor for multiple cell wall enzymes” Cleverley et al. complement previous findings on the cellular roles of GpsB, providing a structural basis for the interaction of the cytoplasmic cell cycle regulator with PBP's and thus provide mechanistic insights in the organization and coordination of the cell wall synthesis machinery. The authors identified the crucial sequence motif, involving a conserved Arg residue in class A PBP's and relevant residues in GpsB for different bacteria. A comprehensive structural analysis of PBP:GpsB pairs using PBP cytoplasmic mini-domains enlightened crucial interaction sites supported by a mutagenesis approach allowing for the detailed binding motif analysis, determination of binding constants and phenotypic analysis. Based on this knowledge additional proteins are predicted to be members of the GpsB interactome. The authors provide a comprehensive set of data and translate structural findings to the cellular level to support their conclusions. The research topic covered in this work is of major importance for the field and highlights the complexities of bacterial cell physiology and the complex enzymatic cell envelope networks. That said, although well written the manuscript is sometimes hard to “digest”. I suggest considering the following points:

Abstract, last sentence: I recommend to delete the statement regarding the design of antibiotics. There is no basis that supports such a statement, nor has this been discussed in the main text. Essentiality (not true for all GpsBs discussed) or being part of a larger protein complex involving essential proteins do not define a “good” antibiotic target. Moreover, in the past such target based approaches largely failed and the development of resistance towards such targets is more than likely.

Author response: Agreed, the final sentence has been removed.

Results, line 107: Can the authors comment on why BsGpsB5-64 has been used for structural analysis and BsGpsB1-68 (wt and variants) for FP and CD? The same is true for SpGpsB4-63 and SpGpsB1-63, respectively or LmPBPA11-20 and LmPBPA11-21.

Author response: In our initial structural studies (Rismondo et al 2016) the terminal few amino acids in GpsB were found to be disordered. Therefore, and only to improve our chances of growing crystals in complex with peptides, we generated new constructs for the production of recombinant proteins for crystallography that lacked a few amino acids at either end, but used the longer constructs for FP and CD. The termini are sufficiently far from the peptide bind site that they play no role. This point has been clarified on lines 110-112, 162-163, 307-311 and 576-578. We apologise for any confusion, which was not our intention.

Regarding *LmPBPA1*, we apologize because a typographic error was made and a *LmPBPA1*₁₋₁₅ construct was used for CD and crystallography experiments whereas the slightly longer *LmPBPA1*₁₋₂₀ peptides were used in FP experiments. We have corrected the manuscript throughout. Regardless,

our BACTH analysis (Figure 2B) indicates that such a difference is not significant. The first 15 amino acids encompass the core region of the *LmPBPA1* cytoplasmic mini-domain that interacts with GpsB because mutations of amino acids downstream of Lys14 do not affect the interaction with GpsB in BACTH.

Line 156: Is BsGpsB1-68 correct in this context (structure elucidation)? Shouldn't it read 5-64?

Author response: Our oversight; many apologies, this typo has been corrected.

Line 150: An alignment of GpsB of the different bacteria used in this study may be helpful. Here it would be especially helpful to highlight GpsB interfacial residues.

Author response: Agreed, this has now been done as suggested to become new Supplementary Figure 1A.

Figure legend 2: Is LmPBPA11-15 correct? The text states (starting line 150): LmPBPA11-21 and LmPBPA11-15. This is very confusing.

Author response: Our oversight; many apologies for this typo and this has now been corrected (see above, and each construct is now defined as it appears).

Line 166: for all relevant amino acids of LmPBPA1 the KD values of respective mutant proteins have been determined, except for Thr7.

Author response: Indeed, but since Arg8 and adjacent backbone atoms observed in the structure of the *LmPBPA1* complex adopts mainchain torsion angles corresponding to an α -helix it is likely that in solution Thr7 N-caps the helix in PBPA1. We have already established with *BsPBP1* that losing the N-cap weakens binding by >7-fold, so it seemed redundant to perform the same experiment with Thr7Ala PBPA1 peptides. We have clarified this point on lines 177-182 and 192-194.

Line 246: exchange “,” to “.”

Author response: Done as requested

Line 255: delete “and GpsB”

Author response: Done as requested

Line 279: self-interaction of GpsB may not indicate functionality.

Author response: We agree that GpsB self-interactions do not guarantee functionality and the sentence has been revised accordingly. What we meant was that if the GpsB allelic variants still self-interacted and interacted with wildtype GpsB (and even with other proteins), this would indicate that all them were expressed, likely properly folded, and able at least to dimerize – since the N-terminal domain of GpsB is an obligate dimer, mutations that abrogate GpsB self-interaction result in a *gpsB* null phenotype (Rismondo et al 2016, Cleverley et al 2016). In addition, the GpsB allelic variants were expressed at wildtype levels, as revealed by Western analysis. We have amended the text on lines 318-323 and 335-340 to clarify this point.

Paragraph starting line 285: Given the enormous interaction studies for Bacillus and Listeria it is noticed that BacTH-analysis is lacking for *S. pneumoniae*.

Author response: BACTH analysis for quite a few cell division proteins has already been reported for *SpGpsB* in the previously-published (and cited) studies by Fleurie et al 2014 and Rued et al 2017. In Rued et al, *SpMreC* and *SpPBP2a* co-immunoprecipitated with *SpGpsB* and genetic experiments suggested that *SpPBP2a* was the class A PBP regulated by *GpsB*. For those reasons we chose to focus first on the interactions between *SpGpsB:SpPBP2a* and *SpGpsB:SpMreC* by BACTH and on the phenotypes of the *gpsB* and *pbp2a* mutants. However, given the complex network of interactions involving *GpsB* detected for *B. subtilis* and *L. monocytogenes*, we used the *Lm* BACTH data as the starting point to ascertain if at least some of these interactions were also conserved in *Sp*. The results of a more extensive BACTH analysis of wild-type *SpGpsB* and its allelic variants are now presented as Supplementary Figure 6A. Notably this identifies an interaction of *SpGpsB* with the essential class B *SpPBP2x* which is also confirmed by new FP analysis (Supplementary Figure 6B) using a peptide encompassing the cytoplasmic mini-domain of *SpPBP2x*. In addition, analysis of the interaction of *SpGpsB* with *SpPBP2b* by BATCH (new Supplementary Figure 6A) and FP (new Supplementary Figure 6B), when combined with previously published co-IP data (Rued et al. 2017), contends that *SpGpsB* and *SpPBP2b* interact indirectly via a bridging protein. We have modified the text accordingly in several places throughout the Results, but most notably in Lines 366-396. Please also see our response to reviewer 1, lines 356-358.

Reviewer #3 (Remarks to the Author):

Cleverley, et al. report the crystal structures of a cell cycle regulator *GpsB* in complex with the cytoplasmic “mini-domains” of PBPs from three bacteria (*B. subtilis*, *L. monocytogenes*, and *S. pneumoniae*). The structures reveal motifs that are important for the protein-protein interactions, leading to identification of two new *GpsB*-interacting proteins in *B. subtilis*. The authors generated extensive amounts of data to analyze the protein-protein interfaces using FP and the bacterial two-hybrid system. They further provide some evidence to show significance of the protein-protein interfaces on the protein functions for cell growth and drug susceptibility. Overall, this is a solid structural biology paper demonstrating detailed interactions of cell wall synthases and the adaptor proteins in the low-GC Gram-positive bacteria.

Specific comments:

1. Are the *BsGpsB* residues important for binding to the PBP1 mini-domain critical for the cellular function of *GpsB*? This can be examined in *B. subtilis* *gpsB ezrA* or *gpsB ftsA* double mutant which shows poor growth with severe cell division.

Author response: We thank the reviewer for this good suggestion. We have not done such an analysis in *B. subtilis* partly because the background strain that the reviewer suggests is already quite sick and we do not wish to run the risk of inadvertently propagating gain-of-function suppressors that could mask the true impact of individual point mutations. We do report in this manuscript the results of a simpler, arguably cleaner analysis in *S. pneumoniae*, in which a *gpsB* deletion is lethal. We hope that the experiments we have conducted in *S. pneumoniae* with mutated versions of *GpsB* that lose binding to PBP2a *in vitro* addresses the reviewer’s question sufficiently: in *S. pneumoniae* the mutation of *GpsB* residues involved in PBP binding including D29A, D33A and Y23A causes similar growth and morphological defects (Figure 5B) as a *gpsB* deletion. These mutations affect neither the expression levels of *GpsB* nor its ability to self-interact (Supplementary Figs 4C and 4D).

2. The affinity of *BsPBP1* to *GpsB* was 0.7 μM in SPR (line 104) and no binding was observed at 25 μM *BsPBP1* lacking the first 17 amino acids, indicating requirement of the 17 aa for the interaction. However, in the FP assay, the affinity of the *BsPBP1*(1-32) peptide was low at 120 μM (Supplementary Table 1). I think this is disconnect and could be due to affinity reduction by the fluorophore attached to PBP1 peptide or due to interactions between other domain of PBP1 and

GpsB. An SPR experiment with the non-labeled PBP1 peptide should clarify this. If the affinity of the peptide is low in the SPR, the submicromolar affinity of GpsB and PBP1 could be an artifact because other PBP1 domain is in TM and the periplasm. In addition, since GpsB is a hexameric protein, what was the stoichiometry binding in the SPR data?

Author response: The higher affinity of full length GpsB for PBP1 in SPR ($0.7 \mu\text{M}$) as compared with FP ($160 \pm 10 \mu\text{M}$; new FP data concerning full length *Bs*GpsB have now been added to Supplementary Table 1) likely reflects an avidity effect resulting from the GpsB hexamer interacting simultaneously with more than one PBP1 molecule on an SPR chip surface. Indeed, the binding data for full-length GpsB indicate a multivalent interaction and only fit to a 1:1 binding model over a limited protein concentration range. It was not possible to determine the stoichiometry of the interaction with certainty using the binding data obtained at higher protein concentrations. Multivalent interaction of the GpsB hexamer with PBP1-coated SRP chips is supported by the much reduced disconnect between binding affinities determined by SPR and FP for *Bs*GpsB₁₋₆₈ (K_d of $40 \mu\text{M}$ and $120 \mu\text{M}$, respectively). Based on our SAXS model of the GpsB hexamer (Cleverley et al 2016) and the crystal structure of the dimeric *Bs*GpsB₁₋₆₈ the maximum distance between PBP1 binding sites is much larger in the GpsB hexamer than in the *Bs*GpsB₁₋₆₈ dimer (up to 90 \AA vs 15 \AA for the hexamer compared with the dimer). Therefore the GpsB hexamer has a greater capacity to engage in multivalent interactions with PBP1-coated SPR chips than the isolated N-terminal domain. Furthermore, in contrast to the GpsB hexamer, the SPR data for the *Bs*GpsB₁₋₆₈ interaction with PBP1 support 1:1 rather than multivalent binding. The influence of avidity effects in SPR measurements of *Bs*GpsB:*Bs*PBP1A interactions has also been discussed by us previously (Rismondo et al. 2016), and we now also cite this work on line 510.

To avoid confusion we have adjusted the text on line 103-108 to focus on the key, important point – SPR chips prepared in the same way but coated with different PBP1 constructs – either with the full length PBP1 or the truncation mutant PBP1₁₇₋₉₁₄ interacts very differently with GpsB and removal of the first 16 amino acids of PBP1A abrogates binding severely.

Regarding the reviewer's suggestion that the fluorophore on the peptide interferes with GpsB binding we were assiduous in choosing to label the peptide at sequence positions distant from the GpsB-peptide interface (as revealed by the crystal structure of the *Bs*GpsB₅₋₆₄:*Bs*PBP1A₁₋₁₇ complex). We have generated *Bs*PBP1A₁₋₃₂ peptides labelled with fluorophores at either residue 16 or residue 31; extensive binding data for 16-labelled peptides is included here while the binding data for the wild type 31-labelled peptide has been previously published (Rismondo et al. 2016) The peptides bind *Bs*GpsB with comparable affinity (K_d of $120 \pm 10 \mu\text{M}$ ([16-labelled] vs K_d of $90 \pm 10 \mu\text{M}$ [31-labelled]), strongly arguing that the fluorophore itself has no impact on the binding interaction. Furthermore, the GpsB-binding affinity of 16- and 31-labelled peptides show the same pattern of sensitivity to R8A, A10P and R28A point mutations (representative binding curves are shown in the figure below for 31-labelled peptides, and can be compared with the 16-labelled

peptides presented in Fig. 1D). All data reported in the original submission are for PBP1 peptides labelled at position 16 because these data fit most reliably to a 1:1 interaction model over a wider protein concentration range. We now refer to the data with 31-labelled peptides in new text to the legend to Figure 1D.

3. In Figure 2C, PBPA1 and GpsB interact each other and may work together, but why is the PBPA1 deletion mutant more resistant to fosfomycin than WT in *L. monocytogenes* while *gpsB*

deletion was more sensitive to the antibiotic? In addition, is the PBP1A PBPA2 double deletion mutant more sensitive to fosfomycin? If so, is the drug sensitivity change of the double mutant comparable to the *pbpA1(Q10P)* PBPA2?

Author response: This is a good question. Fosfomycin acts on MurA, mediating the first committed step of peptidoglycan biosynthesis. Some of us have already published on the increased fosfomycin sensitivity of the *gpsB* mutant and the increased resistance of the *pbpA1* mutant (Rismondo et al., 2017) and mutants lacking *pbpA1* and *pbpA2* are not viable (Rismondo et al., 2015). The increased fosfomycin sensitivity is explained by spatial constraints exerted by GpsB on PBPA1: according to this concept, the increased fosfomycin sensitivity of the *gpsB* mutant is due to unproductive consumption of lipid II precursors through PBPA1 acting in a spatially uncontrolled manner when it is not recruited in a common complex with GpsB in the *gpsB* mutant (as explained in the legend of Fig. 2C). Consequently, we have added additional text to the legend of Fig. 2C.

4. Line 39 in the abstract, the authors state that this could represent a starting point for the design of much needed new antibiotics. However, the manuscript does not include any description or discussion about how this could lead to a starting point to design new antibiotics. So this statement is unclear.

Author response: As both reviewers 2 and 3 have queried this line, the final sentence has been removed.

Minor comment:

The authors used a variety of peptides, proteins, and variants. It would be helpful to show clearly in figures what protein/peptide variants were used.

Author response: These have all been defined more clearly in either the figure or its corresponding legend, as well as in the main text and apologise for any confusion.

REVIEWERS' COMMENTS:

Reviewer #1 (Remarks to the Author):

The authors have adequately and carefully addressed the points that I brought up in my review.

Reviewer #2 (Remarks to the Author):

In the revised manuscript "The cell cycle regulator GpsB functions as cytosolic adaptor for multiple cell wall enzymes" Cleverley et al. addressed issues and concerns raised by the reviewers and provide an improved revision including additional data as requested. In the revised manuscript and rebuttal letter all of my previous criticism and comments are fully answered. In the present form I recommend publication in Nature Communications.

Reviewer #3 (Remarks to the Author):

The authors revised the manuscript with clarifying text and adding data using in vitro FP, the E. coli BACTH system, and co-IP to further show protein-protein interactions, which improved the overall quality of the manuscript. However, the data demonstrated still do not fully support the importance of the GpsB-PBP interactions for cell wall growth and viability. The importance of the N-terminal interacting motifs (or the Arg residues) of GpsB in *S. pneumoniae* was shown in this manuscript. Similar results were shown previously in *L. monocytogenes* (Rismondo et al 2015). The reviewer's suggestion was to add an experiment to show the importance in *B. subtilis*. Authors did not generate the data because the double mutants (*gpsB ezrA* or *gpsB ftsA*) are quite sick. However, another experiment could be designed using either 1) the method used to obtain the results shown in Supplementary Table 2, 2) construction of a *gpsB* down-regulation strain under *ezrA* or *ftsA* deletion background, 3) overexpression of the BsGpsB-binding peptide from PBP1 that would disrupt the GpsB-PBP1 interaction in the *ezrA* or *ftsA* mutant, or 4) overexpression of GpsB mutants that would show dominant-negative effects in the *ezrA* or *ftsA* mutant because GpsB functions as a multimer. Additionally, the results using the *S. pneumoniae* system cannot be translated in the importance in *B. subtilis* sufficiently because the phenotypes of the *gpsB* deletion mutant are different in *B. subtilis* and *S. pneumoniae*. Without the data, the importance of the two key Asp residues of BsGpsB for cell wall synthesis and viability in *B. subtilis* cannot be addressed.

Mutations in the GpsB-interacting motifs of PBPs did not phenocopy *gpsB* deletion and rather showed mild phenotypic defects in *L. monocytogenes* and *S. pneumoniae*. This suggests the GpsB-PBP interaction is not significantly critical for cell wall synthesis and viability and that interactions of GpsB to other proteins may bypass the defects in GpsP-PBP interactions in cells. The authors state about this in line 355-364 and line 447-477. However, this may not match the claim in the abstract (line 30-32) as it can read like the importance of the interactions for cell wall growth and viability. Also with having the additional data in *S. pneumoniae*, the abstract should be revised and updated.

Overall, the paper is of interest for the new protein-peptide complex structures, and

extensive and detailed studies on the protein interactions of GpsB using in vitro FP, the E. coli BACTH assay, and co-IP. I also agree with Reviewer 1 that this manuscript is an extension of previous work published by some of the authors. One issue in the revised manuscript is still that the importance of the GpsB-PBP interactions for peptidoglycan synthesis in the cellular context was not fully addressed, probably due to interactions of GpsB with other proteins that bind PBPs. Because these interacting proteins bind multiple regions of GpsB, the GpsB crystal structures in complex with the PBP peptide exhibit the facet of protein-protein interactions in the GpsB protein network, which should be clearly stated in the manuscript.

Minor point:

The response to Reviewer 3's comment #3 is a reasonable explanation. This should be included in Discussion of the main text rather than in the figure legend because the different phenotypes between the *gpsB* and *pbp* deletion mutants were also observed in *S. pneumoniae* shown in Fig 5D and 5E.

Line 355-366: The BACTH data were generated in *E. coli*, not in *S. pneumoniae*. So, these results only support the critical role for protein-protein interactions, but "do not" support the critical role for the "in vivo" function in *S. pneumoniae* cells.

Line 450-452: As the cytoplasmic mini-domain of PBPs are short and were characterized in the binding assay, it is unclear what are exosites of PBP that enhance the affinity to GpsB directly. It would be more likely that another bridging protein would exist that interacts with cytoplasmic GpsB and the periplasmic or transmembrane domain of PBPs.

REVIEWERS' COMMENTS:

We thank all three reviewers for their time in helping us to improve our manuscript, and we are glad to learn that they found the majority of their critique was addressed to their satisfaction in the revised manuscript. **Our responses to the remaining few issues are detailed below, in red**

Reviewer #1 (Remarks to the Author):

The authors have adequately and carefully addressed the points that I brought up in my review.

Author response: No changes required.

Reviewer #2 (Remarks to the Author):

In the revised manuscript "The cell cycle regulator GpsB functions as cytosolic adaptor for multiple cell wall enzymes" Cleverley et al. addressed issues and concerns raised by the reviewers and provide an improved revision including additional data as requested. In the revised manuscript and rebuttal letter all of my previous criticism and comments are fully answered. In the present form I recommend publication in Nature Communications.

Author response: No changes required.

Reviewer #3 (Remarks to the Author):

The authors revised the manuscript with clarifying text and adding data using *in vitro* FP, the *E. coli* BACTH system, and co-IP to further show protein-protein interactions, which improved the overall quality of the manuscript. However, the data demonstrated still do not fully support the importance of the GpsB-PBP interactions for cell wall growth and viability. The importance of the N-terminal interacting motifs (or the Arg residues) of GpsB in *S. pneumoniae* was shown in this manuscript. Similar results were shown previously in *L. monocytogenes* (Rismondo et al 2015). The reviewer's suggestion was to add an experiment to show the importance in *B. subtilis*. Authors did not generate the data because the double mutants (*gpsB ezrA* or *gpsB ftsA*) are quite sick. However, another experiment could be designed using either 1) the method used to obtain the results shown in Supplementary Table 2, 2) construction of a *gpsB* down-regulation strain under *ezrA* or *ftsA* deletion background, 3) overexpression of the BsGpsB-binding peptide from PBP1 that would disrupt the GpsB-PBP1 interaction in the *ezrA* or *ftsA* mutant, or 4) overexpression of GpsB mutants that would show dominant-negative effects in the *ezrA* or *ftsA* mutant because GpsB functions as a multimer. Additionally, the results using the *S. pneumoniae* system cannot be translated in the importance in *B. subtilis* sufficiently because the phenotypes of the *gpsB* deletion mutant are different in *B. subtilis* and *S. pneumoniae*. Without the data, the importance of the two key Asp residues of BsGpsB for cell wall synthesis and viability in *B. subtilis* cannot be addressed.

Author response: We thank the reviewer again for this idea. We reiterate that we have not performed such an experiment in *B. subtilis* because a phenotype is only evident when *gpsB* is synthetically inactivated with *ftsA* or *ezrA* and these Δ *gpsB/ftsA* and Δ *gpsB/ezrA* background strains are quite sick. A further complication is that all four class A PBPs are dispensable in *B. subtilis*, rendering it extremely difficult to evaluate precisely the direct effect of the GpsB:PBP1 interaction, and its loss when interfacial residues are mutated. This PBP dispensability is quite different to the situation in *L. monocytogenes* and *S. pneumoniae* where we can take advantage of the synthetic lethality of the class A PBP that acts redundantly, to replace the function of the GpsB-binding PBP that is lost when inactivated. We believe that all the data presented herein using different experimental systems *in vitro* and *in vivo* on three bacterial species, and in conjunction with data

published by some of us previously, converge on the critical importance of the conserved aspartates in GpsB function, irrespective of origin. Nonetheless, we have added some additional text in justification of the absence of the proposed experiment on lines 458-467 of the marked-up manuscript.

Mutations in the GpsB-interacting motifs of PBPs did not phenocopy *gpsB* deletion and rather showed mild phenotypic defects in *L. monocytogenes* and *S. pneumoniae*. This suggests the GpsB-PBP interaction is not significantly critical for cell wall synthesis and viability and that interactions of GpsB to other proteins may bypass the defects in GpsP-PBP interactions in cells. The authors state about this in line 355-364 and line 447-477. However, this may not match the claim in the abstract (line 30-32) as it can read like the importance of the interactions for cell wall growth and viability. Also with having the additional data in *S. pneumoniae*, the abstract should be revised and updated.

Author response: We have adjusted the abstract to address this point, in addition to making some slight changes in response to editorial requests.

Overall, the paper is of interest for the new protein-peptide complex structures, and extensive and detailed studies on the protein interactions of GpsB using in vitro FP, the *E. coli* BACTH assay, and co-IP. I also agree with Reviewer 1 that this manuscript is an extension of previous work published by some of the authors. One issue in the revised manuscript is still that the importance of the GpsB-PBP interactions for peptidoglycan synthesis in the cellular context was not fully addressed, probably due to interactions of GpsB with other proteins that bind PBPs. Because these interacting proteins bind multiple regions of GpsB, the GpsB crystal structures in complex with the PBP peptide exhibit the facet of protein-protein interactions in the GpsB protein network, which should be clearly stated in the manuscript.

Author response: We did address this point in the original and revised version of the manuscript (which can be found on lines 479-483 in the marked-up final version of the manuscript) and consequently have made no further adjustments to the text.

Minor point:

The response to Reviewer 3's comment #3 is a reasonable explanation. This should be included in Discussion of the main text rather than in the figure legend because the different phenotypes between the *gpsB* and *pbp* deletion mutants were also observed in *S. pneumoniae* shown in Fig 5D and 5E.

Author response: We have moved the text from the Figure legend to the main text as requested, to lines 229-233.

Line 355-366: The BACTH data were generated in *E. coli*, not in *S. pneumoniae*. So, these results only support the critical role for protein-protein interactions, but “do not” support the critical role for the “in vivo” function in *S. pneumoniae* cells.

Author response: We agree, and consequently have added the following text to line 383-4: “for mediating protein-protein interactions with *SpGpsB*”, and modified the sentence accordingly to clarify this point.

Line 450-452: As the cytoplasmic mini-domain of PBPs are short and were characterized in the binding assay, it is unclear what are exosites of PBP that enhance the affinity to GpsB directly. It would be more likely that another bridging protein would exist that interacts with cytoplasmic

GpsB and the periplasmic or transmembrane domain of PBPs.

Author response: The proposed exosites could be part of the cytoplasmic mini-domain from the PBPs beyond what have been used in vitro (and in crystallo) to determine the binding determinants for GpsB such as the juxta-membrane region, which we had already discussed in the original and revised version of the paper (and can be found on lines 387-392 in the final version of the manuscript). We have added additional text on line 488 in addressing this point. We agree that another protein may be involved as a bridge, and indeed this point has already been discussed (and can be found on lines 243-245, 411-413 and 509-515) and consequently no further adjustments to the text has been made.